# High-dimensional topographic organization of visual features in the primate temporal lobe

Mengna Yao[1,2,4], Bincheng Wen[1,2,3,4], Mingpo Yang[1,4], Jiebin Guo ®[1,4], Haozhou Jiang[1], Chao Feng[1], Yilei Cao[1], Huiguang He ®[2,3] & Le Chang ®[1,2] ✉

The inferotemporal cortex supports our supreme object recognition ability. Numerous studies have been conducted to elucidate the functional organization of this brain area, but there are still important questions that remain unanswered, including how this organization differs between humans and non-human primates. Here, we use deep neural networks trained on object categorization to construct a 25-dimensional space of visual features, and systematically measure the spatial organization of feature preference in both male monkey brains and human brains using fMRI. These feature maps allow us to predict the selectivity of a previously unknown region in monkey brains, which is corroborated by additional fMRI and electrophysiology experiments. These maps also enable quantitative analyses of the topographic organization of the temporal lobe, demonstrating the existence of a pair of orthogonal gradients that differ in spatial scale and revealing significant differences in the functional organization of high-level visual areas between monkey and human brains.

One of the most fundamental questions in neural science is how our brain divides into multiple functional regions. Over the past decades, the organization of one brain area known as inferotemporal cortex (ITC) has attracted lots of attention[1–3]. Located at the top of the ventral visual pathway, this brain region transforms low-level visual features encoded in early visual areas into the concept of objects[4] and provides downstream association areas with necessary ingredients for high-level cognition involving objects, such as object memory[5] and object-related decision making[6]. The close relationship between the neural representation of ITC and the abstract concept of objects, as well as the ease of studying its coding scheme through visual stimulation, have made it an ideal site to examine the functional organization of the brain. A series of studies carried out in primate brains have revealed multiple ITC subregions specialized for specific object categories or features[7–21]. For example, subregions specialized for face processing

have been identified in both human and monkey brains[7,13,22,23]. It has been suggested that these subregions are organized along coarse gradients of visual features such as animacy and object size[1,15,18,24–26]. The large-scale organization of the temporal lobe has also been characterized using data-driven approaches by showing natural stimuli, such as movies, to the subjects[27–29]. These studies suggest the existence of a large-scale visual feature map in the human brain that is consistent across individuals. More recently, theoretical studies have shown that applying simple spatial constraints to the backbones of neural networks can lead to topographic organizations similar to those observed in the experiments, such as face-selective regions[30–34]. Despite all these achievements, some important questions remain unanswered, for example: 1) The brain regions identified with specialized functions cover only about half of the monkey ITC[18], and it's unclear what the rest does; 2) Most studies have been conducted in a

[1]Institute of Neuroscience, Key Laboratory of Primate Neurobiology, CAS Center for Excellence in Brain Science and Intelligence Technology, Chinese Academy of Sciences, Shanghai 200031, China. [2]University of Chinese Academy of Sciences, Beijing 100049, China. [3]Institute of Automation, Chinese Academy of Sciences, Beijing 100190, China. [4]These authors contributed equally: Mengna Yao, Bincheng Wen, Mingpo Yang, Jiebin Guo. ✉e-mail: lechang@ion.ac.cn

single species, either human or a non-human primate species, and a detailed comparison of the ITC's topographic organization between human and monkey brains is still needed. To answer these questions, we need to map out the visual selectivity across the ITC in both species and perform quantitative analyses on the obtained maps.

We set out to tackle these challenges by constructing a high-dimensional object space and compute the selectivity of each brain location within this object space. Inspired by recent advances at the intersection of artificial intelligence and neuroscience[18,35–37], a 25D object space was constructed using responses of units in a deep neural network to a large database of natural images, and functional MRI experiments were conducted in both monkeys and humans to map out the feature preference of the visual temporal lobe. The resulting preference maps helped us determine the functions of previously uncharted territories and reveal differences in the functional organization of high-level visual areas between monkey and human brains.

## Results

### Constructing a high-dimensional space of visual features for fMRI experiments

We aimed to characterize the functional map of the visual temporal lobe using a rich set of visual features. As demonstrated in previous studies, responses of ITC neurons could be accurately predicted by linear combinations of unit activities in convolutional neural networks[35,36]. Therefore, we built the object space by passing 200k natural images from ImageNet, an online database, through AlexNet[38], a neural network for object recognition previously shown to be a good model of ITC neurons[36], and performing independent component analysis (ICA) on responses of units in layer fc7 of this network (Fig. 1a). 25 independent components (ICs) were extracted. By comparing the "signal" values of different IC dimensions for the 200k images, we found that these dimensions were largely independent of each other (Pearson correlation = 0, mutual information < 0.05 for all IC pairs). Representative images with the smallest and the largest angles between their 25D IC coordinates and each IC axis were selected for further experiments (Fig. 1b, c). These two groups of images were termed "positive" and "negative" images of the corresponding IC, respectively. Visual inspection of those images revealed a rich set of features: For some ICs, images were grouped according to object categories (e.g., dogs in the negative images of IC1 and buildings in the positive images of IC25, Fig. 1b, c), while for others, objects with similar shapes and appearances, not necessarily belonging to the same category, were grouped together (e.g., round objects in the positive images of IC5 and stripes in the negative images of IC19, Fig. 1c). Fifty images were selected for each group, resulting in a total of 2500 images (= 50 images×2 groups×25 ICs). Among all these images, there was only one image that appeared more than once, which is much lower than the chance level (= 14.9 images, estimated by 10000 rounds of randomly selecting 25 groups of 100 images from the 200 k images).

In our application of ICA, the data was pre-whitened using principal component analysis (PCA) and reduced to the same number of components as intended by ICA, which were then rotated to maximize non-Gaussianity[39]. As a result, the 25 ICs explained the same amount of variance as the top 25 PCs (78% of the total variance). We did a computational simulation to determine which dimensionality reduction technique should be employed and found that the estimated preferred features of AlexNet units are more robust to random image selections for ICA than PCA (Fig. S1a–c). The decision to use 25 dimensions was inspired by a previous paper showing that 25 linear components of deep-layer unit activations in a convolutional neural network could accurately predict the responses of IT neurons[35]. A recent study also demonstrated that the top 25 PCs, but not higher PCs, of AlexNet activations could be decoded by IT responses[18] (their Extended Data Fig. 11C). Moreover, we found that the 25 ICs of AlexNet fc7 not only

accounted for a large percentage of the total variance in fc7 activations but also explained that of layer fc6 quite well (Fig. S1d–f). As will be described in detail later in the paper, many well-studied visual features, such as curvature and animacy, can be reconstructed by linearly combining the 25 IC axes (see Fig. S4b, c for representative images of the combined axes). To quantify how well these selected images represent the corresponding ICs, the difference between the average IC coordinates of the two groups of representative images was computed. We found that each IC's differential feature was very close to the true IC axis and was insensitive to the specific images being selected (Fig. 1d).

To map out the neural selectivity to these 25 features in the primate brain, we performed functional MRI experiments on three awake macaque monkeys and four human subjects while they fixated on the images. Within each scan, positive and negative representative images of a single IC were presented in alternating blocks. An identical control block of 50 natural images randomly selected from ImageNet, preceded and succeeded by gray-screen blocks, was presented in the middle of each scan (Fig. 1e). Taking advantage of the periodic nature of the stimulus, Fourier analysis was used to extract the sign and magnitude of each voxel in response to the alternation of positive and negative blocks of each IC (Fig. S2a, b), which was later normalized by the response to the control block to facilitate direct comparison across ICs. Only voxels with significantly stronger responses to the control blocks than the gray-screen blocks were selected for further analyses ($p < 0.001$, paired t-test, two-tailed, not corrected for multiple comparisons, see Methods). We performed the normalization using the control block because the entire experiment was conducted over multiple sessions, and the signal strength inevitably varied between sessions. Since we used an identical set of images in the control block, the response to this block could serve as a reference to control for the differential effects of intersession variation on different ICs.

### Consistent neural representation across hemispheres, individuals and species

The normalized response to each IC was then projected onto the flat map of the temporal lobe, where ITC is located, resulting in 25 maps for each hemisphere of each individual. For monkey brains, the boundary between V4 and IT, identified by retinotopic mappings (Fig. 1f), was used to separate the temporal lobe from the occipital lobe[17]. For human brains, where retinotopic areas are more numerous and complex[40,41], anatomical markers (posterior transverse collateral sulcus and the anterior part of lateral occipital sulcus) were used to delineate the temporal lobe. We did not restrict ourselves to ITC alone, because we observed ordered and continuous feature selectivity over the entire temporal lobe. To further enhance the signal quality and facilitate quantitative analyses on the flat map, we resampled the projections with 2 mm × 2 mm squares, by averaging the results of all vertices within each square (Fig. 2a). In these maps, values close to 1 indicate stronger responses to positive images than to negative images, and values close to −1 indicate the opposite. Other square sizes and resampling approaches were also tested, and similar results were obtained for analyses described later in the paper (Fig. S2c–k). A similarity matrix was then computed by correlating the maps of all pairs of ICs (Fig. 2b). As in the classical representational similarity analysis, this matrix captures the way how visual information is represented. By computing the correlations between the similarity matrices, we found that visual features were represented similarly across hemispheres, individuals, and even species (Fig. 2c–f). Furthermore, we found that without the normalization of the control block, the consistency between subjects became lower (Fig. S2l–m), suggesting that this normalization step removed noise from the data.

It should be noted that the consistency between representational similarities is expected from the stable tuning of high-level visual areas as demonstrated in previous studies[18,23,42], and the purpose of this

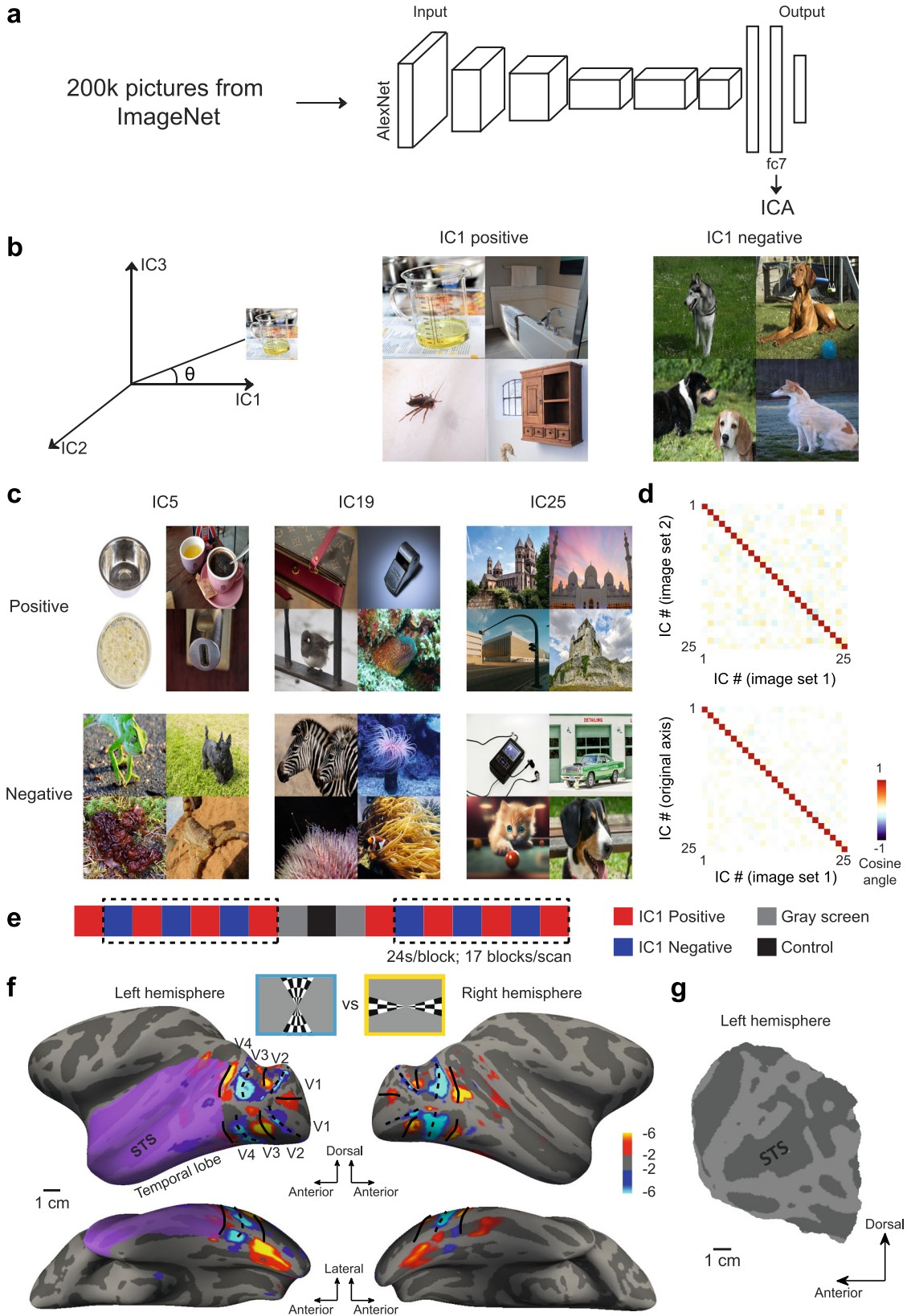

analysis is to illustrate the reliability of our procedure. It's also important to note that our feature maps include not only the ITC, but also the dorsal part of the temporal lobe, which is mainly involved in processing visual motion and auditory information. Further analyses suggest that the estimated preferred features of the dorsal areas are reliable and distinct from those of the ITC (Fig. S3).

## Explaining the feature preference map using interpretable visual features

To comprehensively examine the feature selectivity of different temporal lobe regions, 25 feature maps were merged into one single 25D feature map, with feature preference at each location represented by a 25D vector (termed the "preferred feature" of that location, Fig. 3a).

**Fig. 1 | The construction of 25D object space and stimuli for fMRI experiments.**
**a** 200 k natural images from ImageNet, an online database, were passed through
AlexNet, a neural network for object recognition. Independent component analysis
was performed on the responses of units in the penultimate layer (fc7). 25 inde-
pendent components (ICs) were extracted to build an object space. **b** For each IC,
the angles between 25D vectors of 200k images and the IC axis were computed, and
50 images with the smallest and the largest angles were selected to represent the
positive and negative ends of that IC, respectively. Representative images for IC1
are shown on the right. **c** Representative images for more ICs. **d** The difference
between the average 25D vectors of the positive and negative representative ima-
ges was computed for each IC. Besides the 50 most extreme images (image set 1),
images with the 50th to the 100th smallest and largest angles were selected as a
separate image set (image set 2). The cosine angles between the differential vectors

estimated using two image sets and those between the differential vectors and the
original IC axes are shown in the upper and lower panels, respectively. **e** A typical
stimulus sequence for the fMRI experiment. In each block, 48 images were ran-
domly selected from the 50 representative images and presented in random order.
The dashed lines indicate the blocks used in the following analyses. **f** Boundaries of
the temporal lobe and retinotopic visual areas were identified using retinotopic
mapping. The visual stimuli are shown above the inflated surfaces of one monkey.
Two different perspectives of the inflated surfaces are shown. The color scale bar
indicates the common logarithm of the probability of error. STS, superior temporal
sulcus. **g** The flat map of the left temporal lobe of the same monkey (purple patches
in **f**). Note that due to copyright restrictions, the original ImageNet images in (**b**, **c**)
are replaced with natural images from Pixabay (https://pixabay.com/).

Each 25D preferred feature is an axis in the 25D "object" space (usually
a combination of multiple IC dimensions, see Fig. S4h–k) that can be
used to characterize the neural tuning at the corresponding location[18].
In order to understand which aspects of visual information are repre-
sented by these axes, we compared them with a collection of inter-
pretable visual features, including those investigated in previous
studies. The term "visual feature" is used to refer to a label or an index
of an image that reflects some property of that image. In total, 21
features were examined, including: low-level features, such as energy
at specific spatial frequencies and color; mid-level shape features, such
as curvature; high-level semantic features, such as animacy. To directly
compare these "visual features" with the "neural features" (the 25D
preferred feature measured experimentally), we projected them into
the 25D IC space (Fig. 3b). Typically, this projection was done by first
averaging the corresponding values of the visual feature across images
within "positive" and "negative" blocks of different ICs, and then
computing the difference between the average feature values of two
blocks for each IC and normalizing the 25D vector to unit length (for
details, see Methods: The construction of interpretable features and
comparison with neural data; see also Fig. S4 for visualizations of the
projected 25D features; For the relationships between the 21 features,
see Fig. S4d) and compared to the neural features measured experi-
mentally (Fig. 3b). The squared cosine angles (SCA) between neural
and visual features were used to quantify how well each interpretable
visual feature explained the neural data. This allowed us to quantita-
tively link our feature maps to already known features and potentially
identify novel features represented in the temporal lobe. Note that the
squared cosine angles between any neural feature and a set of com-
plete orthogonal features add up to 1 (Fig. 3c, inset). We use the terms
"low-", "mid-" and "high-level" only to help the reader intuitively
understand what the features are—addressing the level of neural
representation requires different stimulus sets and experimental
designs[43,44]. Examining the squared cosine angles, we can see that
some of the 21 features explain the responses much better than other
features (Fig. 3c).

To identify the set of most explanatory features, we designed a
feature selection procedure by adding one feature at a time to
optimize the total explained proportion of neural responses at each
step (see Methods: feature selection). We used a half-split approach
to determine the optimal number of features: the 25D feature of one
location estimated using half the data was fitted by linearly com-
bining the selected features, and the feature of the other half was
then compared to the fit (Fig. 3d). We found that seven features
performed the best (Fig. 3f, black line and arrow; see Methods: fea-
ture selection). The seven features are (red arrows in Fig. 3c): 1) mean
magnitude of spatial frequency between 1.45 cyc/degree and 2.91
cyc/degree (named "high spatial frequency" or "high sf"); 2) exis-
tence of animals in the picture (named "animacy"); 3) existence of
humans in the picture (named "human"); 4) a stubby-spiky axis
estimated by projecting object images with stubby or spiky shape
(named "stubby-spiky"); 5) mean magnitude of spatial frequency

between 0.18 cyc/degree and 0.36 cyc/degree (named "low spatial
frequency"); 6) curvilinear/rectilinear shape; 7) PC1 of AlexNet fc6
(roughly stubby/spiky shape, see Fig. S4c and Bao et al.[18]). Interest-
ingly, we found these seven features explained only ~56% of the
neural responses after being normalized by the noise ceiling, sug-
gesting that some unknown features are encoded by the primate
temporal lobe. Next, we extracted additional features from the
neural data, by first orthogonalizing the neural features with respect
to the top seven features, and then performing principal component
(PCA) analysis on the orthogonalized features. Whenever PCA was
employed in the following text, the dimensionality of the original
data was always that of the object space ( = 25), so each PC repre-
sented an axis in the 25D space. Adding two PCs to the seven features
performed the best in explaining the neural features, achieving 82.5%
of the noise ceiling (purple line and arrow). These two features were
termed residual features 1 and 2. The number of most explanatory
features used to orthogonalize the neural features does not strongly
affect the directions of the two residual features (Fig. S5b).

We then compared these nine features with two types of baseline
models (Fig. 3e): 1) PCs of AlexNet responses; 2) WordNet labels of the
representative images. ImageNet is organized according to the
WordNet hierarchy, so each image in the database is associated with
multiple labels, e.g., a dog could be labeled as a "hunting dog", which is
subordinate to higher-order labels such as "domestic animals".
Twenty-one labels were screened based on the number of repre-
sentative images related to each label (no less than 100 and no more
than 2000 out of all 2500 images are required). Squared cosine angles
between these labels and neural features were considerably lower than
those for the 21 interpretable features (Fig. S5c; cf. Fig. 3c). The most
explanatory features were selected out of the 21 labels using the same
procedure described above. Overall, the two types of baseline models
performed much worse than the selected interpretable features when
the number of features were matched (compare blue and orange lines
to the black and purple lines in Fig. 3f).

Next, we examined the spatial distribution of the 9 features on the
flat map (Fig. 4a and Fig. S5d). Feature 1, the "high spatial frequency"
feature, largely avoided the superior temporal sulcus (STS, its fundus
was indicated with dotted lines in Fig. 4a), showing strong similarity to
neural features in the ventral ITC and the superior temporal lobe;
Feature 2, the "animacy" feature, was very prominent along 2-3 stripes
around and orthogonal to the fundus of STS. Other features exhibited
diverse spatial patterns. For example, Feature 6 was positively corre-
lated with the anterior temporal lobe (curvilinear region), and nega-
tively correlated with the ventral ITC (rectilinear region); Feature 8, the
residual feature 1, was negatively correlated with the STS region,
except from the "animacy" stripes identified in Feature 2's map. The
spatial relationship between functionally defined subregions can be
better appreciated by assigning different colors to regions with dif-
ferent preferences (Fig. 4b), with red indicating "high spatial fre-
quency", blue indicating "animacy", and green indicating residual
feature 1. These three features were selected due to their strong

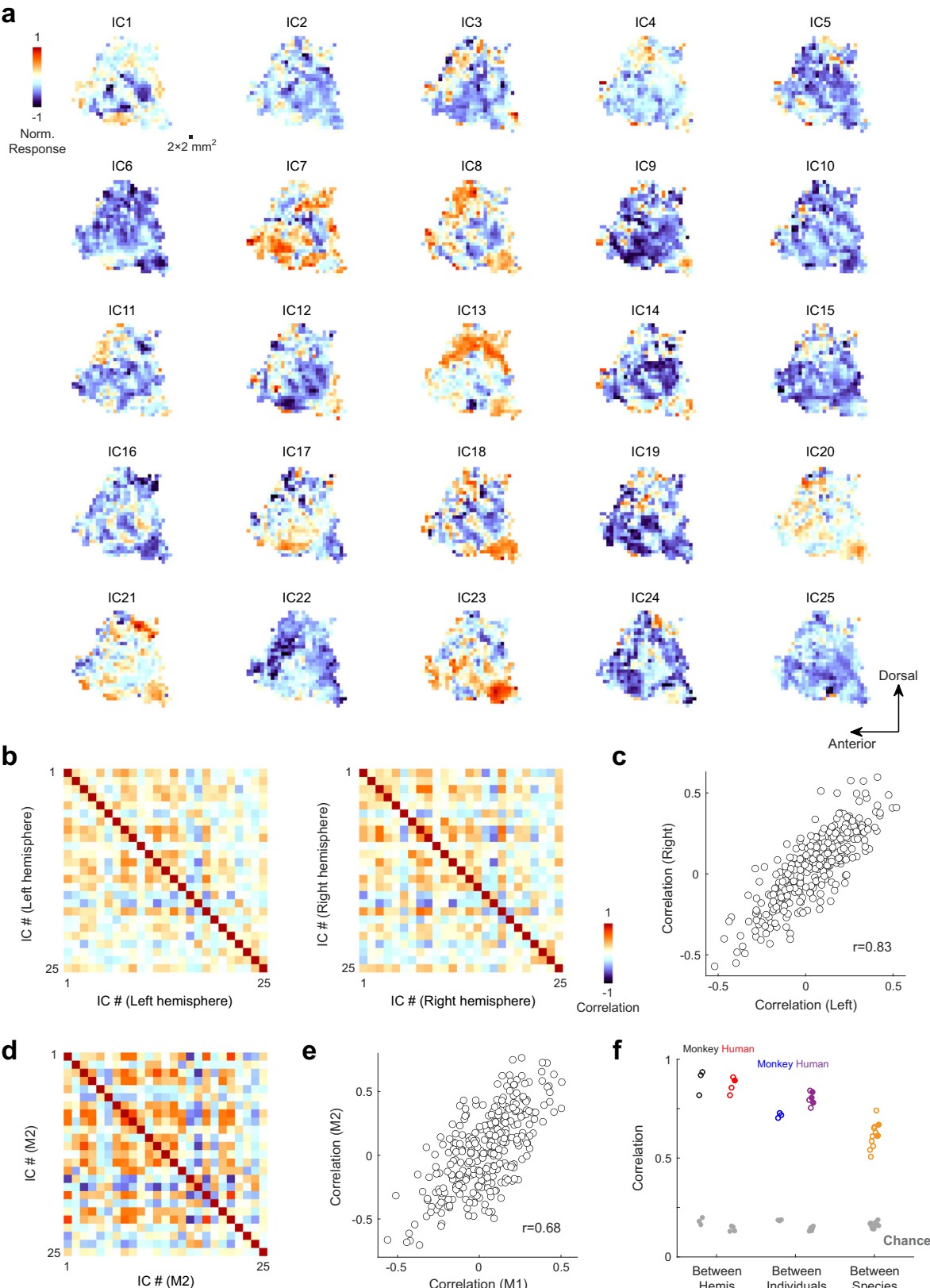

explanatory power for neural features (see annotations in Fig. 4a). We saw alternating blue and green stripes around STS, surrounded by red regions for an individual monkey (Fig. 4b). While previous studies highlighted feature gradients orthogonal to the fundus of STS, roughly along the dorsal-ventral axis[2,18,45], our results suggest the existence of a periodic gradient of feature preference parallel to the STS fundus.

Here, we used the term "gradient" to indicate the gradual change in the similarity between the 25D preferred features and a given dimension across the cortical surface (Fig. S7m).

To assess the consistency of the results between animals, three monkeys' brains were registered to a common template of macaque brains (see Methods). We found the spatial organization of feature

**Fig. 2 | Consistent neural representation of 25 IC axes across hemispheres, individuals and species. a** 25 feature maps of the left temporal lobe for one monkey (M1), plotted on the cortical flat map. Results of all vertices within a 2 mm × 2 mm square were averaged. Each map shows the preference of all locations in the temporal lobe for the positive representative images of one IC axis over the negative ones, with the largest absolute value normalized to 1. In these maps, values close to 1 indicate stronger responses to positive images than to negative images, and values close to -1 indicate the opposite. **b** Population similarity matrices in two hemispheres. The correlation coefficients between pairs of feature maps were computed as a 25 by 25 matrix. **c** Correlation coefficients for one hemisphere were compared to those for the other. **d** Population similarity matrix of a second monkey. **e** Correlation coefficients for monkey 1 were compared to those for monkey 2. **f** Consistency of similarity matrices, measured by Pearson correlation, between hemispheres (3 monkeys and 4 humans), individuals (3 monkey pairs and 6 human pairs), and species (3 × 4 monkey-human pairs). For interhemispheric consistency, open and solid dots indicate male and female subjects, respectively. For inter-subject consistency, open and solid dots indicate within-gender and between-gender comparisons, respectively. Gray dots indicate upper bounds of 5% confidence interval based on 1000 times of random shuffling. Source data are provided as a Source Data file.

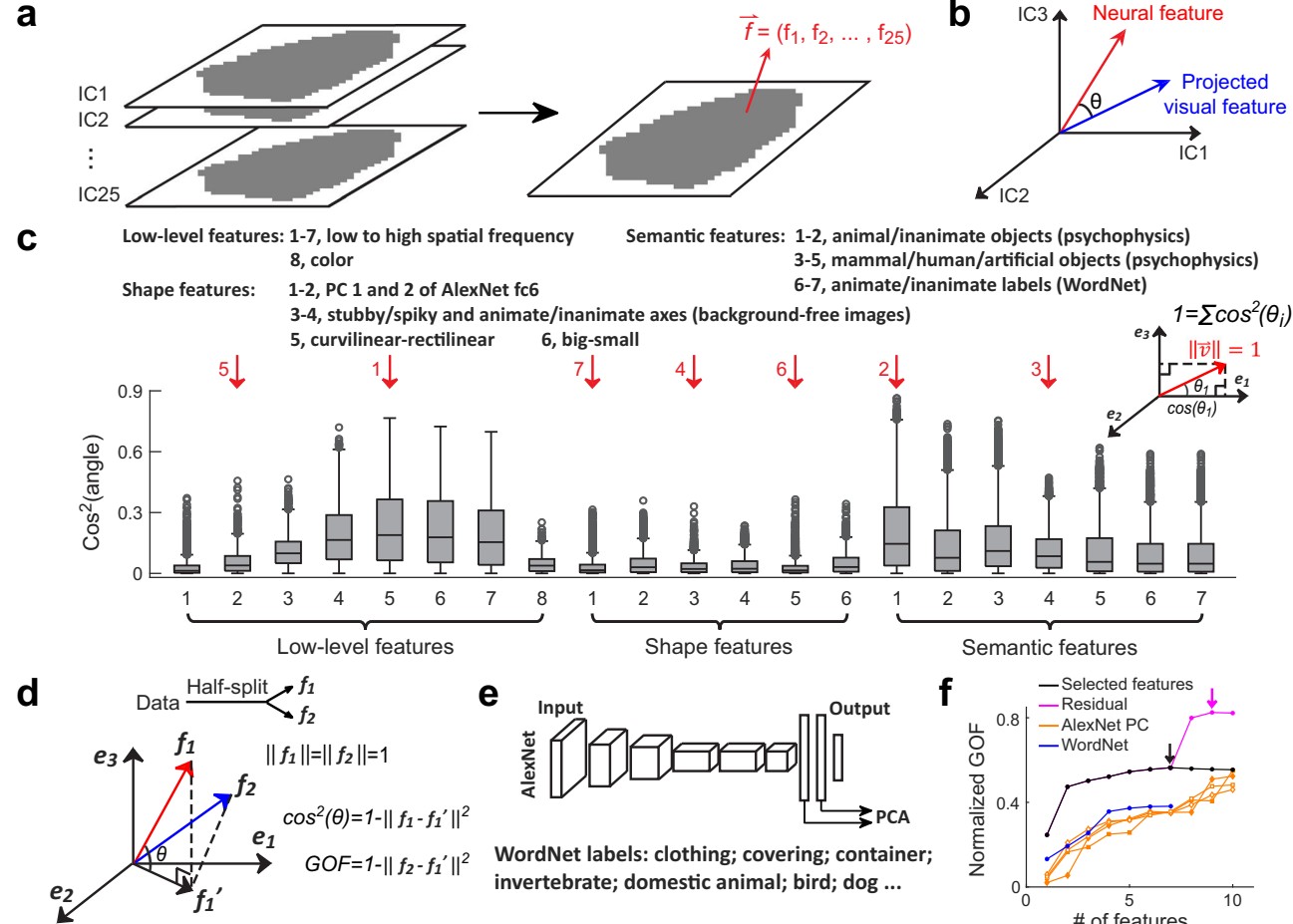

**Fig. 3 | Explaining 25D feature maps using interpretable visual features. a** The 25 feature maps in Fig. 2 were merged into a 25D feature map, where each location's preferred feature was represented by a 25D vector. **b** 21 visual features were projected into the 25D object space (blue) and compared with the neural features (red). **c** Similarity between 25D neural features of three monkeys and 21 features (see Methods). The squared cosine angle between two 25D feature vectors was used to quantify their similarity (inset). Box plots show the median (line), quartiles (boxes), range (whiskers), and outliers (circles); $n = 3321$ brain locations from three monkeys. **d**–**f** Feature selection. A set of most explanatory features were selected by adding one feature at a time to optimize the total explained proportion of neural responses at each step (see Methods). **d** The procedure for quantifying the explanatory power of a set of features (see Methods). $f_1$ and $f_2$ indicate the 25D preferred features estimated using two halves of the data; $e_1$ and $e_2$ indicate the selected features. **e** Two baseline models. 1) Principal components of the responses of AlexNet units in four layers (fc6 and fc7, before and after ReLU) to 200k ImageNet images to construct the 25D object space. 2) WordNet labels. Each selected label was required to contain no less than 100 and no more than 2000 out of all 2500 representative images (see Fig. S5c). **f** Goodness-of-fit, normalized by the noise ceiling (see Methods), is plotted against the number of features for multiple models. The model in (**c**) is denoted by the black line, with the arrow indicating the optimal feature number ( = 7). Additional features were derived by first orthogonalizing the neural features with respect to the selected seven features, and then performing PCA on the orthogonalized features. Incorporating these orthogonal features continued to increase the goodness-of-fit (purple line), until 9 features were selected (purple arrow). Orange lines indicate PCs of AlexNet units, with different symbols representing different layers. The blue line indicates the WordNet labels, which underwent the same feature selection process as the model in (**c**). Source data are provided as a Source Data file.

preferences was largely consistent across individuals (Fig. S4l–m). When projected onto the three features mentioned above, the population-averaged map displayed largely the same pattern as the individual monkey (Fig. 4d). We found a roughly similar pattern in the spatial distribution of feature preference in the human brain: alternating blue and green regions are neighbored by red areas on the ventral side (Fig. 4c, e; see Fig. S6 for the location of the flat map on the inflated surface). However, the topographic organization seems to be less structured in the human brain (compare 4b with 4c, 4d with 4e).

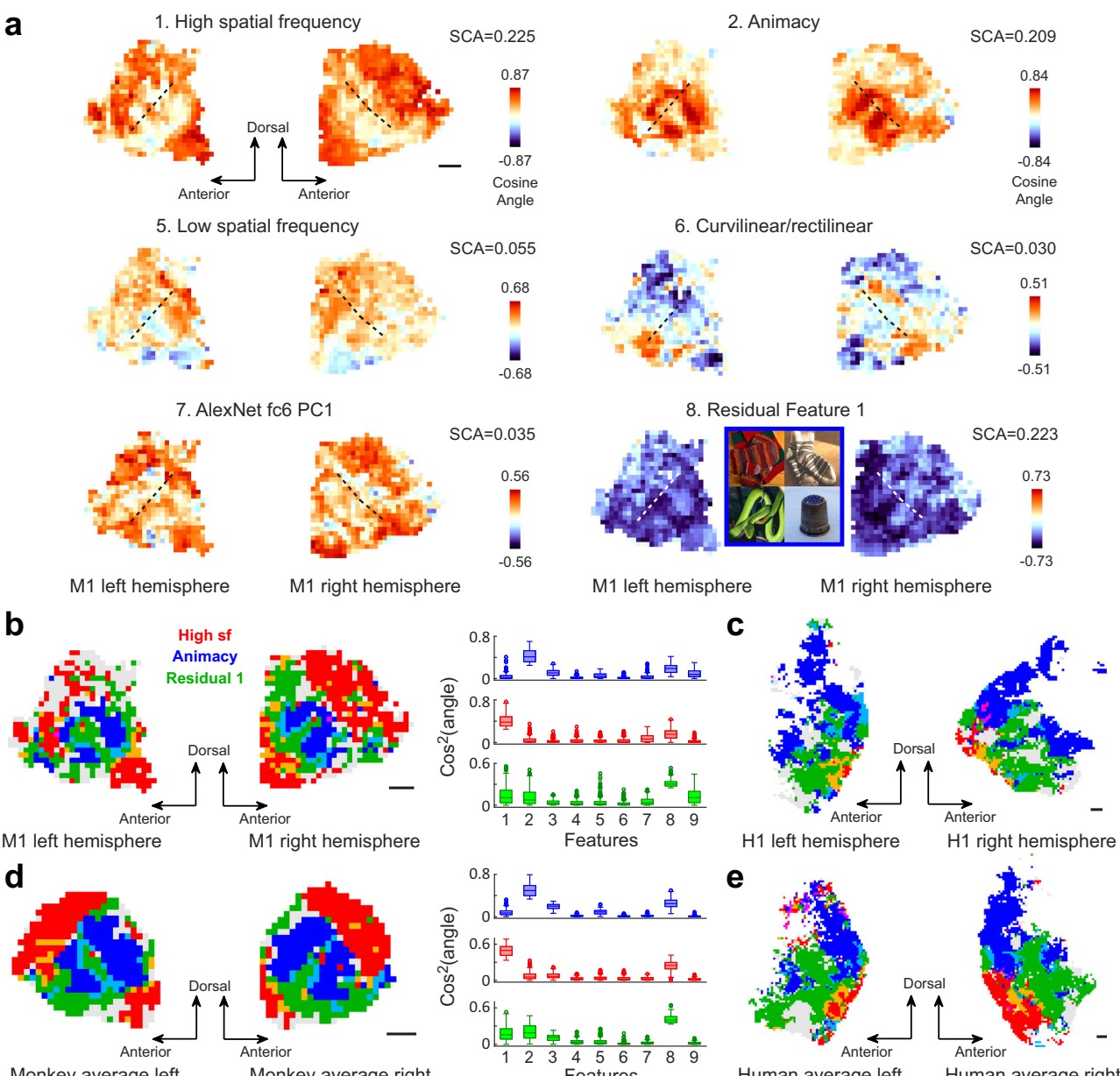

**Fig. 4 | Topographic organization of interpretable visual features. a** The 25D feature map of monkey 1 was projected onto the 9 selected features. Dashed lines indicate the STS fundus. Mean squared cosine angles (SCA) of each feature were shown on top of the scale bar. Negative representative images for residual feature 1 are shown in the inset. Note that the original ImageNet images are replaced with natural images from Pixabay (https://pixabay.com/). Due to space limitations, three feature maps are shown in Figure S5d. **b** Within a single map, different colors are assigned to squares with different preferences: blue, red, and green represent preferences for high sf, animacy, and residual feature 1, respectively. Regions preferring more than one feature are indicated by mixing primary colors (yellow=red+green, cyan=blue+green, purple=blue+red). A square will be labeled if the cosine angle between its neural feature and three features of interest is larger than 0.50 (in the case of residual feature 1, cosine angle < −0.50 was required since most squares were negatively related to this feature). This threshold was chosen because

the overlapping squares of the three regions amount to 10% of all squares. The same criterion applies to (**c−e**). Note that gray squares also belong to the feature map—they are not colored because their similarities with the three features are not strong enough. Squared cosine angles between features of each colored region and the 9 selected features are shown on the right. Box plots show the median (line), quartiles (boxes), range (whiskers), and outliers (circles). From top to bottom: $n = 193, 399$, and $345$ brain locations from one monkey, respectively. **c** Same as (**b**), but for a human subject. The threshold for cosine angle is 0.28. **d** Same as (**b**), but for the average result of three monkeys projected onto the monkey template. The threshold for cosine angle is 0.58. From top to bottom: $n = 271, 293$, and $271$ brain locations, respectively. **e** Same as (**d**), but for the average result of four human subjects projected onto the human template. The threshold for cosine angle is 0.41. Scale bars: 1 cm. Source data are provided as a Source Data file.

It's worth noting that our findings do not imply that the primate temporal lobe is composed of only three regions, each of which prefers one of the three features. We didn't plot other features in the maps of Fig. 4b, d because they didn't show up when the same threshold for the three strongest features was applied (Fig. S5f, h). However, consistent patterns could be observed when a lower threshold was used ( = 1/3 of

the original threshold, Fig. S5g, i). Moreover, when we performed a Gaussian Mixture model on our preference map for clustering, 10-20 clusters were found for each monkey based on Bayesian information criterion (Fig. 5). In addition, visualization of these clusters by their representative images intuitively matched the expectations for face patches[13] and scene areas[16] (contours in Fig. S5j).

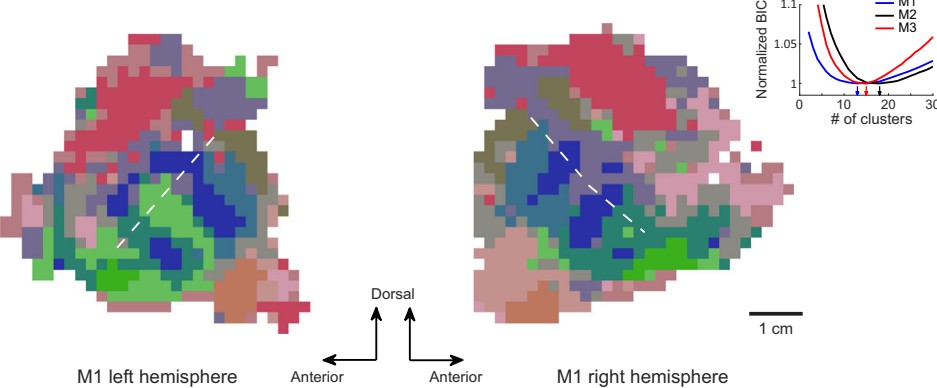

**Fig. 5 | The monkey temporal lobe consists of 10-20 spatially organized functional clusters.** Gaussian mixture model was used on the 25D feature map for clustering. Bayesian information criterion revealed an optimal number of clusters between 10 and 20 for three monkeys (inset). The clustering result for M1 is shown. Locations belonging to different clusters are labeled with different colors, derived from an MDS analysis (R, G, B = the 1st, the 2nd, the 3rd MDS dimension). The white dashed lines indicate the fundus of STS. Source data are provided as a Source Data file.

## Testing new predictions based on the 25D preference map

Next, can we make predictions based on the feature preference map that were not known in previous studies and test them with further experiments? One recent paper found that monkey ITC consists of four networks, coding four quadrants of a 2-dimensional object space[18]. These two dimensions were PCs extracted from layer fc6 of AlexNet, roughly corresponding to "stubby shape vs. spiky shape" and "animate vs. inanimate" axes, therefore the four networks were named spiky-animate(SPA), stubby-animate(STA), spiky-inanimate(SPI) and stubby-inanimate(STI) respectively, with SPA and STA corresponding to body and face patches in past literatures[13,14]. We performed a similar experiment to localize the four networks in two monkeys. Overlaid on the map of the "animacy" feature (cf. Fig. 4a), we found that the two networks representing animate objects, SPA and STA, overlapped nicely with the "animacy" stripes around STS, consistent with the fact that they are concerned with animals (Fig. 6a). The two networks representing inanimate objects, SPI and STI, were located more ventrally than the animate regions (Fig. 6a), consistent with the direction of a coarse animate-inanimate gradient in the primate temporal lobe[3,18]. It is easy to see that the aforementioned four networks are unable to account for the gaps between the "animacy" stripes in the vicinity of STS (i.e., green stripes in Fig. 4b, d). We found the 1st principal component of feature maps of all three monkeys showed a consistent pattern, with positive scores around STS (Fig. S7a), so we named the region with positive PC1 scores the "peri-STS region". The preferred feature for these gaps within the peri-STS region, termed "inanimate stripes", could be estimated using the 25D feature map. This new feature axis explained 26% of neural responses in the full feature map, and was most closely related with the 1st residual feature (or feature 8) among the 9 features selected in Fig. 3 (Fig. S5e).

Positive and negative representative images were then selected from the 200 k natural images in ImageNet for this new feature, with positive images typically containing objects with fine textures (Fig. 6b, see also Fig. S4g for preferred images synthesized for that feature). We performed a separate fMRI experiment using the two groups of images, and found the regions with significant bias towards positive images fell outside of the "animacy" stripes around the STS fundus (white outlines in Fig. 6a), consistent with our prediction. Furthermore, within the peri-STS region, t-contrasts between the two conditions were positively correlated with the projections of corresponding 25D features onto the preferred feature of the inanimate stripes, which was used to select the images, and negatively correlated with the projections onto the "animacy" feature, consistent with predictions based on the 25D feature map (Fig. 6c; The 1% confidence interval of Pearson

correlation was estimated by 2000 iterations of random sampling with replacement from all the scans).

To further confirm our finding, we performed single-unit recordings in one of the new regions localized by fMRI (Fig. 6d). We found consistent neural selectivity at the population level, with overall stronger responses to the "texture" images in the positive block (Fig. 6e and the purple line in Fig. 6h; The top purple line in Fig. 6h indicates the time window with significant difference between the neurons' t-contrasts and 0, $p < 0.01$, Student's t-test, two-tailed, $n = 86$ cells). As a control, we recorded in a region of the same monkey that responded more strongly to the negative block in the fMRI experiment (Fig. 6f) and found a different pattern there, with stronger responses to the images in the negative block (Fig. 6g and the green line in Fig. 6h; the top green and black lines in Fig. 6h indicate the time windows with significant difference between the negative region's t-contrasts and 0 and between the two regions' t-contrasts, respectively, $p < 0.01$, Student's t-test, two-tailed, $n = 56$ and 86 cells in the negative and positive regions, respectively). Furthermore, we found the preferred features of neurons in these two regions, estimated by the same 25-IC stimulus set used in the fMRI experiment, could be clearly distinguished from each other along the 1st principal component of all neurons (Fig. 6i; In this case, the firing rate for each stimulus was measured with a time window of 50–400 ms after stimulus onset), suggesting a division of labor in processing visual features between the two areas. Finally, in addition to the representative images, we also presented the synthesized images for the same feature to the monkey (Fig. S4g), and found that neurons in the new region (Fig. 6d) responded more strongly to the synthesized images of the positive direction of that feature than to those of the negative direction (Fig. 6j and the brown line in Fig. 6k). Interestingly, the positive synthesized images elicited even stronger responses than the positive representative images selected from ImageNet (black line in Fig. 6k), likely because the synthesized images are better aligned with the target feature. While we used the term "texture" to describe the function of the inanimate stripe, we are also aware that this is more or less a qualitative description. Partially due to the diversity of naturalistic textures, it is difficult to replace this description with a simple equation. On the other hand, the 25D preferred feature is a quantitative characterization itself, from which we know that this area performs previously unknown computations.

In sum, our results not only confirmed the existence of a new area, but also supported the presence of a fine gradient along the less-explored posterior-anterior direction (see the rightmost panel of Fig. S7a for their relative locations), which was thought to be largely related to the hierarchical processing within ITC, e.g., building invariance to viewing angles[18].

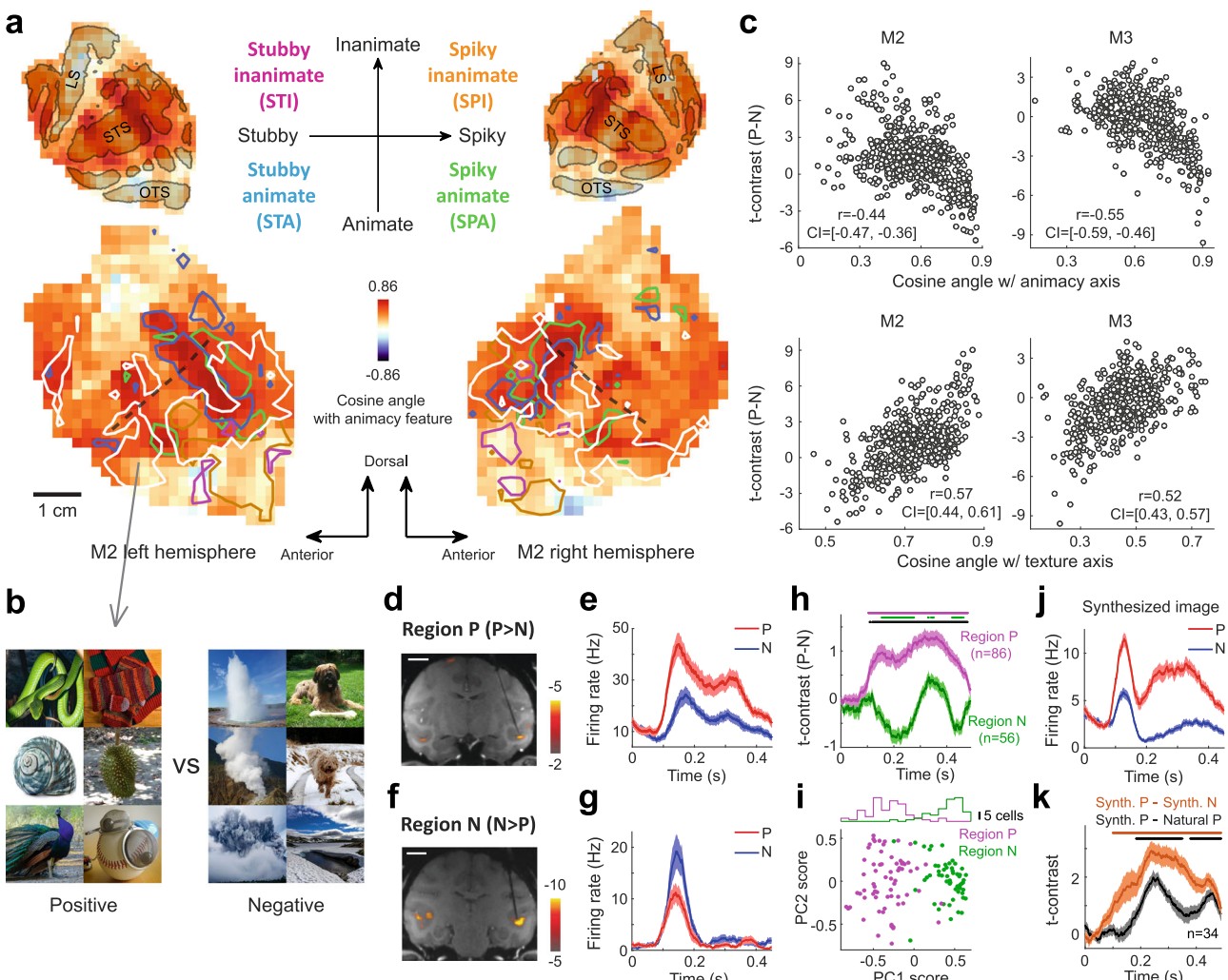

**Fig. 6 | Validating the functional selectivity of a new ITC subregion. a, b** Four functional networks reported in a previous paper[18] were localized in the temporal lobe of one monkey, shown as contour plots (STA and SPA: $p < 10^{-4}$; STI and SPI: $p < 10^{-6}$; Student's t-test, two-tailed, not corrected for multiple comparisons; the color code is shown in the inset) overlaid on the map of the animacy feature (**a**). Preferred (positive) and nonpreferred (negative) images were selected for sub-regions within the peri-STS region left out by the four networks (**b**). Note that the original ImageNet images are replaced with natural images from Pixabay (https://pixabay.com/). A separate experiment comparing positive to negative images revealed new subregions in the peri-STS region (white outlines in **a**, $p < 0.001$). Dashed lines indicate the STS fundus. The locations of main sulci are shown above (LS: lateral sulcus; OTS: occipitotemporal sulcus). **c** In two monkeys, t-contrasts between the two conditions were negatively correlated with projections of 25D features onto "animacy" feature within the peri-STS region, and positively correlated with the projections onto the feature dimension used to select the images. CI: 1% confidence interval. **d** A coronal slice showing the location of a positive-block

preferred region in M3 targeted for electrophysiology. Scale bar indicates the common logarithm of the probability of error. **e** Responses of an example neuron to images in positive and negative blocks of the fMRI experiment (see Methods). Shadings represent SEM. **f, g** Same as (**d, e**), but for a negative-block preferred region. **h** Average time courses of t-contrasts between the two conditions, with purple and green lines indicating positive- and negative-block preferred regions, respectively ($n = 86$ and 56 cells). Shadings represent SEM. Top lines indicate significant time windows ($p < 0.01$, Student's t-test, two-tailed, see text). **i** PCA was performed on the 25D preferred features of all neurons recorded in two areas, estimated using the 25-IC stimulus set. The distributions of PC1 scores are shown above. **j** An example neuron's responses to the synthesized images of the same feature in (**b**) (Fig. S4g). **k** Average time courses of t-contrasts between positive and negative conditions for the synthesized images (brown) and those between positive conditions for the synthesized and natural images (black). Same convention as (**h**). $n = 34$ cells. Source data are provided as a Source Data file.

Another prediction relates to the difference in feature preferences between the two primate species. When PCA was conducted on the preferred features of all individuals pooled together, a clear difference between two species was observed on the 2nd PC, with much higher scores in monkeys than in human subjects (Fig. 7a). We found the 2nd PC could be well approximated by linearly combining "animacy" feature and "high sf" feature (see Methods: Comparing feature preferences of human and monkey brains and the bottom-right corner of Fig. 7a). When examining the animate region and the inanimate region separately, we found the feature preference of the inanimate region differed strongly between the two species (Fig. 7b). While the inanimate region of the human brain showed a preference largely opposite

to the "animacy" feature (this is an indication of its preference for inanimate objects, since the cosine angle between the "animacy" feature and the "inanimate object" feature is −0.96), that of the monkey brain showed little selectivity for the "animacy" feature, but instead strongly preferred the "high sf" feature (Fig. 7b, bottom-left and bottom-middle). These differences result in opposite tunings along the dimension combining the two features between the two species (Fig. 7b, bottom-right). Positive and negative representative images of the combined dimension were then selected−consistent with the meaning of the two features, positive images typically contained animals within crowded environments, while largely isolated artificial objects were found in negative images (Fig. 7c). A separate experiment

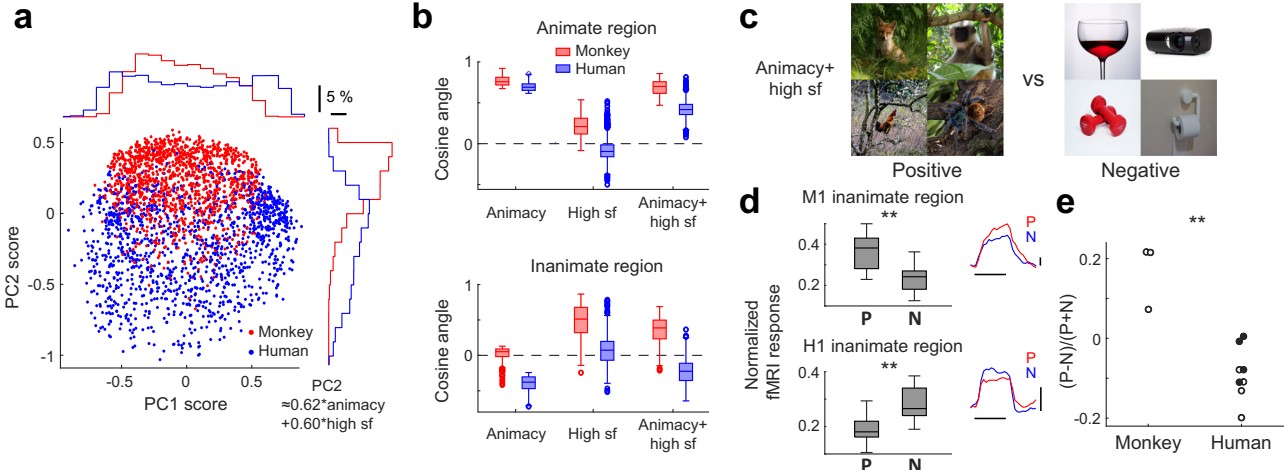

**Fig. 7 | Comparison of feature preferences between the two primate species.**
**a** PCA was performed on the preferred features of all subjects pooled together. To balance the contributions of the two species, the human data (blue) were down-sampled to the size of the monkey data (red). Only 40% of all data points are plotted to avoid overcrowding. Marginal distributions of the full data set are shown.
**b** Cosine angles between preferred features of the animate/inanimate regions in the two species and three features, including: "animacy" feature, "high sf" feature, and sum of the two. Feature preferences of all squares were projected onto the "animacy" feature, and squares with top/bottom 15% projections were defined as the animate/inanimate region. Box plots show the median (line), quartiles (boxes), range (whiskers), and outliers (circles); $n = 498$ and 2041 brain locations from 3 monkeys and 4 human subjects, respectively. **c** Positive and negative representative images of the combined feature (animacy+high sf). Note that the original ImageNet images are replaced with natural images from Pixabay (https://pixabay.com/).
**d** Normalized fMRI responses of the inanimate region to the two conditions for one monkey and one human subject (see Methods). Box plots show the median (line), quartiles (boxes), range (whiskers), and outliers (circles). The results from multiple scans for the two conditions were compared using a paired t-test (two-tailed, $p = 2 \times 10^{-6}$, t(10) = 9.52, $n = 11$ scans for monkey M1, $p = 2 \times 10^{-4}$, t(11) = −5.59, $n = 12$ scans for human subject H1). Since in this experiment we recruited human subjects who were not characterized for the 25D feature map, the inanimate regions were determined as the 15% squares with the largest t-contrasts between inanimate and animate blocks of the four-object-type stimuli (Fig. 6a). The average response time courses to two stimulus conditions are shown in the insets. Horizontal bars indicate stimulus duration ( = 24 s). Vertical bars represent a 1% change in fMRI signal. **e** Comparison of the preference for positive over negative representative images between human and monkey inanimate regions. Each dot represents one individual. $p = 4 \times 10^{-4}$ between the two groups (Student's t-test, two-tailed; t(9) = 5.42, $n = 3$ monkeys and 8 human subjects). Open and solid red dots indicate male and female subjects, respectively. Source data are provided as a Source Data file.

using the two groups of images confirmed our prediction, with significantly stronger bias towards the positive condition in monkeys than humans (Fig. 7d, e).

So far, our analyses have focused on elucidating which visual features were represented in different subregions of the temporal lobe. We then examined the feature preference map from a different perspective, by asking: What general principle underlies the topographic organization of visual features on the cortical surface? While this issue has been studied in the past by characterizing the spatial relationship between discrete subdomains, our largely continuous high-dimensional feature map allowed us to adopt a more global approach. We viewed the 25D feature map as the superposition of 25 feature gradients, one gradient for each IC, and searched for the organizing principles of visual features by quantitatively analyzing these gradients. It's noteworthy that these 25 maps were not independent of each other (Fig. 2b), suggesting that they probably share similar structures. One way to discover such shared structures is principal component analysis (PCA). In the following, we will first discuss the implication of PCA's results on the dimensionality of the map ( ≈ the number of independent gradients), and then move on to analyze the detailed topographic organization for each PC.

**Organizational principles of visual features in the primate temporal lobe**
Previously, topographic maps characterized on the two-dimensional cortical surface were typically based on feature spaces with one or two dimensions[15,18,24,25,46]. To identify the dimensionality of our feature map, PCA was performed on the neural data (Fig. 8a). To test the statistical significance of each PC dimension, a half-split approach was employed (see the left panel of Fig. 8b and Methods for details). We found the feature maps of all three animals possessed no less than

10 significant dimensions (Fig. 8b, right). The same held true for human subjects (Fig. S7b). A simulation based on the neural data suggests our approach hardly overestimates the dimensionality of the feature map (Fig. S7c–f), confirming that the actual feature map in the primate temporal lobe is of high dimensionality in nature. We then examined how this high-dimensional map is organized on the two-dimensional cortical surface.

Diverse patterns were observed when the spatial distribution of feature preference for each PC was examined (Fig. 8c, the column on the right illustrates how each PC and the 9 interpretable features relate to each other as axes in the 25D space): PC1 scores showed a coarse periodic gradient orthogonal to the STS fundus; PC2 scores were negative in the "animacy" stripes orthogonal to the STS fundus (cf. Fig. 4a, the 2nd panel), and positive in the rest of the peri-STS region, displaying a fine periodic gradient parallel to the STS fundus; PC3 scores were dissimilar to either PC1 or PC2 scores. We quantified the spatial patterns by computing spatial autocorrelations on each PC, a method commonly used to describe the presence of systematic spatial variation of a variable defined as a function of space (see Methods: 2D spatial autocorrelation). We found that PC2 displayed a much finer periodic pattern than PC1, with PC3 in the middle (Fig. 8d). This observation was quantified by performing Fourier analysis on autocorrelation maps along two orthogonal orientations (Fig. 8e), with PC1 and PC2 displaying clear orientation selectivity at low and high frequencies (arrows in the right panel of Fig. 8e), respectively. Repeating Fourier analysis along all different orientations, a polar plot was generated for each PC, with angle denoting orientation and radius denoting frequency. We found different patterns at different PCs (Fig. 8f), but consistent patterns across animals (Fig. 8g–i). In particular, feature gradients at PC1 and PC2 were largely orthogonal to each other, dominating at low and high frequencies, respectively (Fig. 8i).

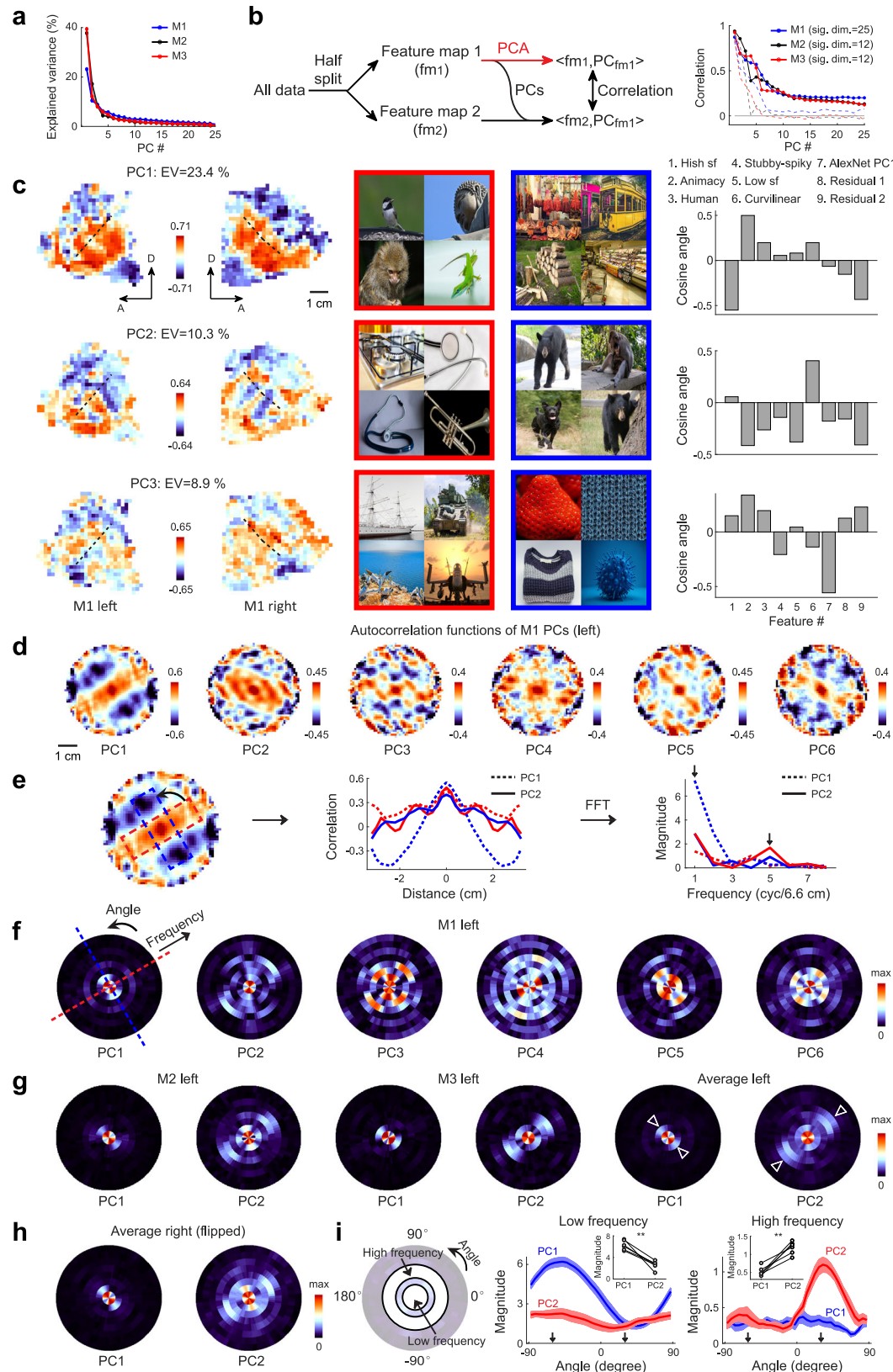

While we have mentioned the pair of orthogonal gradients, one roughly along the dorsal-ventral axis and the other along the posterior-anterior axis, earlier in the paper (e.g., see Fig. 4b, d), here we are able to quantitatively depict their spatial characteristics. Since preference maps for PCs are linear combinations of 25 IC maps that were measured experimentally, we further validated our results on the original

IC maps. We found the low-frequency magnitude of each IC was positively correlated with its absolute loading for PC1, but not for PC2 (Fig. S7g, i), whereas the high-frequency magnitude showed the opposite pattern (Fig. S7h, j).

These findings suggest the map of visual features in the temporal lobe can be viewed as a superposition of multiple periodic feature

**Fig. 8 | High-dimensional topographic organization of the primate temporal lobe. a** Eigenvalues for principal components (PCs) of monkey feature maps. **b** Left: Procedures for half-split analyses (see Methods). Right: Solid lines indicate the average correlations of 500 iterations of half-splitting; dashed lines indicate the lower bounds of 1% confidence interval. **c** Projections of the 25D feature map onto the top three PCs (left), the top PCs' positive and negative representative images (middle; note that the original ImageNet images are replaced with natural images from https://pixabay.com/), and cosine angles between the top three PCs and the 9 features in Figs. 3, 4 (right). **d** Spatial autocorrelations of the top six PCs for M1's left hemisphere. **e** Autocorrelations within two orthogonal rectangles (left) were projected onto their long edges (middle). Fourier analyses were then performed on the projections (right). **f** By rotating the rectangles, the magnitude of Fourier transform was extracted for every angle and frequency, resulting in a polar plot for each PC. **g** Results for the left hemispheres of two other monkeys and the average of three monkeys for PC1 and PC2 of M1's feature map, the same two dimensions as in (**c**–**f**).

Triangles in the population-averaged graphs indicate preferred orientations for PC1's low frequency and PC2's high frequency. **h** Average results for the right hemispheres of three monkeys. The plots are flipped right to left to match the left hemisphere plots. **i** Magnitudes in the polar plots for PC1 (blue line) and PC2 (red line) are plotted against polar angles. For symmetry reasons, only angles in the range [−90°, 90°] were considered. Results for low and high frequencies are shown separately in the middle and on the right (low frequency=average of 1 and 2 cycs/6.6 cm, high frequency = average of 4 and 5 cycs/6.6 cm; see left). The two orthogonal orientations in panel **e** (arrows) roughly matched the tuning peak for PC1 at low frequency and that for PC2 at high frequency. Shadings represent SEM ($n = 6$ hemispheres). Insets: Peak magnitudes for PC1 and PC2 were compared (paired t-test, two-tailed, $p = 0.001$ and $t(5) = 6.63$ for low frequency, $p = 9 \times 10^{-4}$ and $t(5) = -7.03$ for high frequency, $n = 6$ hemispheres). Source data are provided as a Source Data file.

gradients, varying in both spatial frequency and orientations (Fig. S8a). This perspective offers an explanation for why some feature/category occupies multiple subregions—these are peaks or troughs of a fine periodic gradient (Fig. S8b), and is also compatible with the existence of coarse gradients in the primate brain. We do not intend to claim that this new perspective is more fundamental than the conventional view, in which different features/categories occupy different subregions of ITC, since its biological basis is not clear. However, it did allow us to characterize the topographic organization more quantitatively than previous studies, paving the way for more in-depth investigations into the biological mechanisms underlying its formation.

A potential pitfall of this analysis is that the 2D autocorrelation map was computed using the relative locations on the flat map, which might not accurately represent the original spatial relationships on the curved cortical surface in 3D. For instance, in an extreme case, the differential preference for low/high frequency along the two orthogonal orientations (red and blue rectangles in Fig. 8e) might result from differential distortion during surface flattening. We therefore examined the relationship between geodesic distance, the shortest distance between two locations on the curved cortical surface, and the straight-line distance on the flat map, and found no clear difference between those two orientations (Fig. S7k–l), ruling out this possibility.

## Comparing the organizational patterns of visual features between monkey, human, and alternative models

Our analyses suggest the monkey temporal lobe has an intricate organization of visual features, where the spatial scales of feature gradients depend on their orientation. However, since this high-dimensional feature map is embedded in the 2D cortical surface, it seems expected that different feature dimensions will display different spatial patterns, and a likely solution is to represent different features with different oriented gradients. If this is the case, the reader may ask: are there other alternative solutions under the constraint of spatial continuity? In this section, we will quantitatively compare primate feature maps with several alternative models.

While conventional deep neural networks didn't take into account the spatial relationship between units in deep layers, several recent studies have developed network models that incorporate such spatial relationships and encourage nearby units to have similar response profiles[30–34]. After training, response correlation between units decreased monotonously as a function of their distance, and classical ITC subdomains like face patches could be observed. We first implemented one of these models to test whether the functional organization observed in the monkey temporal lobe could arise naturally in the model[30]. Since the previously proposed model was intended to mimic the inferotemporal cortex, we removed the superior part of the temporal lobe in our monkey feature map (shown in grayscale in Fig. 9a). Inspired by a previous study demonstrating the importance of cortical

shape on the topographic patterns that can form within it[47], we used the shape of the map to arrange the units in deep layers. Three network architectures were tested, using either one layer (L6 or L7) or two layers (L6 and L7) to mimic ITC (Fig. 9f). We also used seven functions to constrain how response similarity varies with the distance between units, with two functional forms (reciprocal function and Laplacian of Gaussian; shown as solid and dashed lines, respectively, in Fig. 9g) and multiple widths. Each model was trained with three different random seeds. Altogether, 63 network models were trained (for more details, see Methods: Topographic neural network model). We found the response correlation of units in "ITC" layers varied as a function of the distance between them, in a way consistent with the spatial constraint (Fig. 9h; in the right panel, only networks with the maximum weighting on the spatial loss are taken into account, see Methods), and functional subdomains, such as face patches, could be observed (Fig. 9i).

To systematically compare network results with monkey data, we performed the following analyses to quantify the distinct pattern of topographic organization in ITC. We first computed the spatial autocorrelation for each IC map (Fig. 9b, c), then fitted the map with sinusoidal functions at different frequencies and orientations. Goodness-of-fit, quantified by explained variances, were then averaged across ICs and illustrated as a polar plot (Fig. 9d, e; only the right half of the plot is shown due to central symmetry). In monkeys' data, two clear peaks appeared at different frequencies and nearly orthogonal orientations (triangles in Fig. 9e, corresponding sinusoidal functions are shown to the right), a pattern consistent with the pair of orthogonal gradients shown in Fig. 8 (e.g., see triangles in the last two panels of Fig. 8g), whereas in network models, the distribution of explained variances appeared narrower in terms of frequency, with little orientation selectivity (Fig. 9k). These observations were further corroborated by computing the following indices based on the polar plots: preferred frequency, frequency tuning width, orientation selectivity, and orthogonality index (Fig. 9l–o; see Methods: 2D spatial autocorrelation). The values of the latter three indices would be high if a pair of orthogonal gradients was clearly present in the feature map—e.g., the orthogonality index would be highest if explained variances at two separate frequencies showed strong preferences for orthogonal orientations. We found these three indices varied with the preferred frequency across the networks, and their values were higher in three monkeys than in all the network models with comparable preferred frequencies (Fig. 9p, q, red and gray dots). These results suggest that the previously proposed model could replicate some of the known topographic organization of ITC, such as face patches, but they are not able to reproduce the pair of orthogonal gradients observed in our study.

Furthermore, we constructed two other types of alternative models and found they could not reproduce the pattern in the monkey data as well: 1) self-organizing map models[31,34] were constructed using 25D IC coordinates as inputs (Fig. S9a–c); 2) the monkey feature map

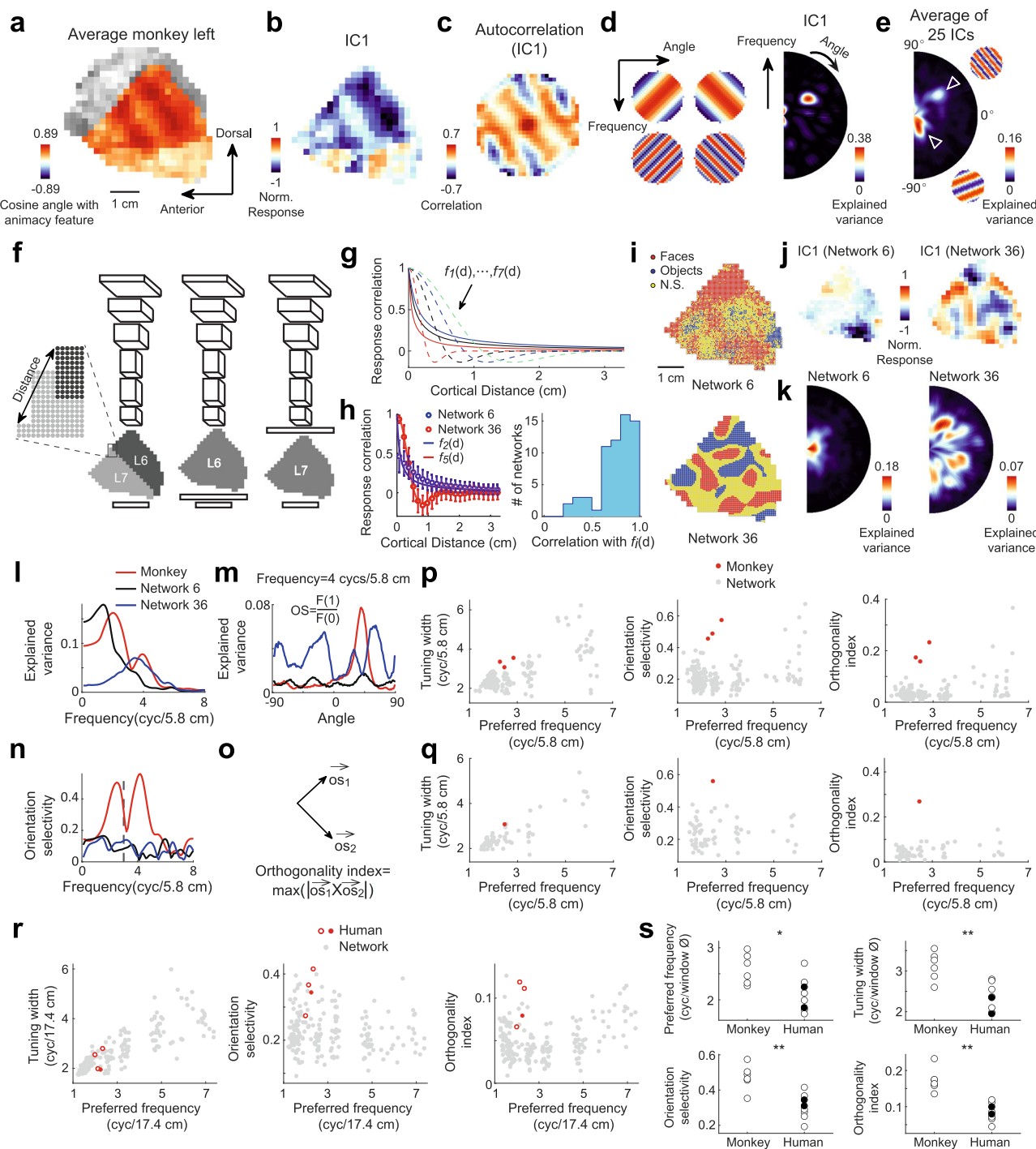

was shuffled to match one of the aforementioned constraining functions (Fig. S9d–f). Our findings suggest that the intricate spatial organization observed in the monkey brain is not an inevitable result of embedding a high dimensional map into the 2D cortical surface under the constraint of spatial continuity. Note that since we couldn't test all possible models, it is likely that the pattern in the monkey data could emerge under some other constraints, but we think identifying such constraints is beyond the scope of the current paper.

We also examined the human temporal lobe using the same set of indices, which were compared to neural network models with units arranged according to the shape of the human feature map (Fig. S9g–i). Interestingly, we found that human data were largely indistinguishable from the models (Fig. 9r). By directly comparing human and monkey data, we confirmed the intricate spatial

organization of visual features observed in the primate temporal lobe, embodied in a pair of orthogonal gradients with different spatial scales, was more evident in monkeys than in humans (Fig. 9s and Fig. S9j–l). We think this difference is likely due to our need to recognize various types of artificial objects, such as tools and letters[10,12], which leads to the reconfiguration and distortion of the originally ordered topographic organization in the brains of our ancestors during evolution.

## Discussion

Over the past decades, the functional organization of high-level visual areas in the primate brain has been intensively investigated. Past studies have identified several networks specialized for processing a particular type of object or feature[7–21], revealed the large-scale

**Fig. 9 | Comparison of feature maps between primates and alternative models.** **a** Cosine angles between average features of three monkeys and the animacy feature, plotted on the left temporal lobe of the monkey template. **b**, **c** IC1's feature map (**b**) and autocorrelation (**c**). **d**, **e** Each IC's autocorrelation map was fitted with sinusoids of different frequencies and orientations. Explained variances, shown as a polar plot (**d**), were averaged across ICs (**e**). **f** Network architectures (see text). **g** Seven functions were used to constrain how response correlation between units vary with cortical distance. **h** Left, Relationship between response correlation and cortical distance for units in two trained networks (bin = 1.5 mm; $n$ = 222828, 607829, 936397, 1220800, 1508523, 1632486, 1835337, 1938684, 2099738, 2109536, 2080679, 2052299, 2002141, 1932663, 1830959, 1676084, 1564828, 1389092, 1201783, 1040034, 841442, and 659991 unit pairs in each bin for network 6; $n$ = 49004, 156206, 231122, 309040, 374379, 427758, 449747, 503386, 506777, 527946, 509935, 520644, 487729, 494298, 444406, 437258, 377210, 359787, 303565, 253240, 212334, and 166229 unit pairs for network 36). Dots: mean; error bars: SD. Curved lines represent corresponding constraining functions. Right, Distribution of the correlation between response correlations and values of the constraining function of cortical distances. **i** Face- and object-selective units in two networks. **j** Feature maps of two networks for IC1 (see Methods). **k** Same as (**e**), but for two networks. **l** Following (**e**) and (**k**), maximum explained variance across orientations is plotted against frequency for the monkey map and two networks. **m** Explained variance is plotted against orientation for a specific frequency. **n** Orientation selectivity (see Methods) is plotted against frequency. **o** Definition of orthogonality index (see Methods). **p** Frequency tuning width, orientation selectivity, and orthogonality index are plotted against preferred frequency for networks and monkeys. Red and gray dots represent monkeys and networks. **q** Same as (**p**), but for the average map of the same network trained three times or three monkeys. **r** Same as (**p**), but for human subjects and corresponding networks. Open and solid red dots indicate males and females, respectively. **s** Comparing monkeys and human subjects using four indices in (**l**–**o**). Each dot represents one hemisphere. Open and solid red dots indicate males and females. Student's t-test (two-tailed) was performed for each index: $p$ = 0.012, 0.001, 0.001, and $10^{-4}$; t(12) = 2.96, 4.18, 4.37, and 5.82; $n$ = 6 and 8 hemispheres for monkey and human subjects, respectively. Source data are provided as a Source Data file.

## Table 1 | Cross-species comparison

| Species<br>Property | Monkey | Human |
|---|---|---|
| Feature selectivity | Similarity:<br>Animate regions show strong preferences for animals<br>Difference:<br>Inanimate regions show weak preferences for inanimate objects and strong preferences for energy at high spatial frequency | Similarity:<br>Animate regions show strong preferences for animals<br>Difference:<br>Inanimate regions show strong preferences for inanimate objects and weak preference for energy at high spatial frequency |
| Topographic organization | Similarity:<br>High-dimensional feature map; alternating "animacy" and "residual" feature preferred regions are neighbored by "high sf" preferred region<br>Difference:<br>A pair of orthogonal gradients with different spatial scales is clearly present | Similarity:<br>High-dimensional feature map; alternating "animacy" and "residual" feature preferred regions are neighbored by "high sf" preferred region<br>Difference:<br>No clear evidence for orthogonal gradients |

organization of high-level visual areas[27–29], and proposed general principles underlying such organization[1,15,18,24–26]. However, several important questions remain unanswered (see Introduction). In this paper, we constructed a high-dimensional space of visual features using a deep network previously shown to be a good model of high-level visual neurons, and systematically characterized the functional feature map in both monkey and human brains. We made the following major discoveries: 1) a new functional subdomain encoding a previously unknown feature was identified in the monkey temporal lobe (Figs. 3, 4, 6); 2) the monkey feature map consisted of a pair of orthogonal gradients with different spatial scales, which were significantly less salient in the human brain (Figs. 8, 9; Table 1); 3) the human brain shows a stronger preference for inanimate objects than the monkey brain (Fig. 7; Table 1).

With the rapid advances of machine learning, convolutional neural networks (CNNs) have been shown to be powerful tools for understanding the primate visual system[35,36]. Although the features we used are extracted from deep neural networks (DNN) but not from neural responses, the recent evidence that DNN features accurately predict the responses of visual neurons[36,37,48] and our decision to use a large number of features at the same time allowed us to 1) obtain a reasonably good approximation of the true neural tuning; 2) reveal the topographic organization of the primate temporal lobe in detail. By analyzing the neural tuning using features derived from AlexNet, what we have done is like projecting the true neural tuning into a subspace spanned by AlexNet features (Fig. S1g). Although AlexNet has been shown to be an excellent model of high-level visual areas[18,36,49], there is still likely to be a significant fraction of neural responses that cannot be explained by these features. Future work is needed to explore additional features to approach the true tuning functions. In addition to the features of static 2D images, visual information not represented in the

ImageNet images, such as binocular disparity and temporal dynamic information, should also be considered.

While previous studies have identified several color selective regions in ITC, we didn't see a strong tuning to color in our dataset. We think it's likely due to the difference in the stimulus sets used. Previous studies have typically compared color images to grayscale versions of the same images, such as gratings, to localize color areas[50,51]. Compared to the conventional color localizer, our stimulus set consisted mostly of natural images, which are much richer in the object shape and category (as these images were collected for the purpose of testing object categorization ability), but may not cover a similar range of color variations (as color is intentionally manipulated in the conventional localizer), favoring tuning to non-color dimensions in our experiment.

Functional subdomains in ITC have been shown to consist of three parts[18], consistent with the periodic pattern along the posterior-anterior axis in some dimensions of our feature map (such as IC1 in Fig. 2a). However, this pattern was not observed for other dimensions (such as IC17). We think this is due to the richness of the stimulus set we used, e.g., IC17 contains isolated objects in the positive block and crowded scenes in the negative block (Fig. S1h), which is not typical for conventional functional localizers.

Our design of sampling the negative and positive ends of each IC dimension was based on the assumption that IT neurons linearly encode axes of high-dimensional spaces spanned by DNN features[18,36]. In the future, multiple locations along a single axis could be sampled to reveal the full tuning along that axis, which would better reveal the feature gradients for individual dimensions.

One central quest of neuroscience is to understand how our brain divides into multiple functionally specialized subregions. Due to the convenience of generating and presenting visual stimuli, topographic

organizations of low-level visual features, such as the orientation of moving bars, have been intensively studied in early visual areas[52]. The large-scale organization of high-level visual areas have also been characterized in several earlier studies[1,15,18,24–29]. In our work, quantitative analyses have been conducted on the global organization of the high-dimensional feature map of the primate temporal lobe, allowing for accurate comparisons across different species or systems. In Fig. 8, the autocorrelation analysis was performed on the projection maps derived from PCA. Previous studies have suggested that PCA may favor the presence of periodic patterns in individual components[53], but note that similar results can be obtained by directly analyzing original feature maps without performing PCA (Fig. S7g–j; Fig. 9). To intuitively understand the difference between monkey brains and alternative models, we compared their feature maps at several steps of the analyses (Fig. S10). We found that for the network models, each gradient is periodic but not clearly oriented, with grid-like autocorrelations; different gradients occupy roughly the same spatial scale but are not aligned in orientation. These observations are inconsistent with the pair of orthogonal gradients at different spatial scales in the monkey map.

While several previous studies compared high-level visual areas in humans and monkeys[23,42,54–58], most of them focused on one or a few semantically defined functional subregions, such as face patches. By examining the largely continuous feature map of the visual temporal lobe, we identified two previously unknown differences between the two species (Table 1; see also Fig. S11 for more comparisons): 1) the human brain shows a stronger preference for inanimate objects than the monkey brain (Fig. 7b–e); 2) the topographic organization of the monkey brain is more regular than the human brain (Fig. 9s, Fig. S9j–l). These two differences are likely due to our need to recognize various types of artificial objects and symbols[10,12], which might expand the cortical areas responsible for inanimate objects and distort the originally ordered topographic organization in the monkey brain during evolution.

## Methods

### Experimental model and subject details

Three adult rhesus macaques (*Macaca mulatta*, all males, 4–6 years old, weighing 5–8 kg) were used in this study. All experimental procedures were approved by the Biomedical Research Ethics Committee of the Institute of Neuroscience, Chinese Academy of Sciences, and were in accordance with the National Institutes of Health Guide for the Care and Use of Laboratory Animals.

Eighteen healthy human subjects participated in the psychophysical (four males and five females, aged between 23 and 26 years) and fMRI experiments (five males and four females, aged between 23 and 38 years). All experimental protocols were approved by the Biomedical Research Ethics Committee of the Institute of Neuroscience, Chinese Academy of Sciences. All subjects had given written consent to the procedure in accordance with institutional guidelines and the Declaration of Helsinki. Subjects were compensated for their participation in the experiment: 1 Chinese yuan/minute for the psychophysical experiment and 2 Chinese yuan/minute for the fMRI experiment. Sex/gender was not considered in the study design and, as a result, the number of subjects is insufficient for sex/gender-related analyses. Sex of participants was determined based on self-report.

### Visual stimuli

**Selection of representative images for 25D feature space.** We built a 25D object space by passing 200 k natural images from ImageNet, an online database, through AlexNet[38], a neural network for object recognition whose deep-layer units were shown to nicely explain the responses of ITC neurons[36], and performing independent component analysis (ICA) on the responses of units in layer fc7[39]. The 200 k images span 1000 object classes, with 200 images randomly selected from

each class. We used unit activations before ReLU because a larger portion of the response variance could be explained by the same number of dimensions, e.g., 25 independent components (ICs) could account for 78.1% of the total variance in fc7 responses before ReLU but only 49.5% after ReLU. For each IC, 50 representative images out of the 200k images with the smallest and largest angles between their 25D IC coordinates and the IC axis were selected (0° ≤ angle between vectors ≤ 180°), with the two groups of images termed "positive" and "negative" representative images of the corresponding IC, respectively (Fig. 1; note that due to copyright restrictions, the original ImageNet images in all figures are replaced with natural images from Pixabay: https://pixabay.com/). Each scan consisted of "positive" and "negative" image blocks from a single IC, presented in alternation. Besides the representative images for ICs, 50 images were randomly selected as a set of control images and were presented in all fMRI sessions to serve as a baseline. For a typical block sequence of the fMRI experiment, see Fig. 1e. Each image had a size of 11° × 11° visual angles and was displayed for 500 ms. Unless otherwise stated, each fMRI scan consisted of seventeen 24-s blocks and lasted 408 s. In each block, 48 images were randomly selected from the 50 representative images and presented in random order. Each scan consisted of 8 blocks of positive images and 6 blocks of negative images. The same number of positive and negative blocks were used in the analysis (= 6 blocks), as indicated by the dashed lines in the current Fig. 1e. Three monkeys (all males) and four human subjects (three males and one female) were scanned for this stimulus set. For each IC, 12.9 and 7.7 scans were collected on average for each monkey and human subject, respectively (17.7, 9.2, and 11.6 scans per IC for individual monkeys; 8.8, 8.4, 8.0, and 5.5 scans per IC for individual human subjects).

**Localizer for face and scene selective areas.** The localizer for face patches contained five types of blocks, consisting of faces, hands, technological objects, vegetables/fruits and bodies. Face blocks were presented in alternation with non-face blocks. The face block was repeated four times and each of the non-face blocks was shown once. A block of gray-screen preceded each stimulus block and was added at the end of each scan. Each scan lasted 408 s.

The localizer for scene areas included two types of blocks: scene block and object block. The scene block contained images of indoor and outdoor scenes, while the object block contained images of faces, bodies, technological objects, vegetables, and fruits. Two types of blocks were presented in alternation. Each scan lasted 408 s.

**Localizer for four object types.** Stubby/spiky animate and stubby/spiky inanimate regions were localized using stimuli similar to those from a previous paper[18]. The full stimulus set contained four types of blocks, consisting of stubby animate (faces), spiky animate (animal bodies), stubby inanimate, and spiky inanimate objects. In each scan, the four blocks were each presented twice, interleaved with blocks of gray-screen. Each scan lasted 408 s.

**Meridian mapping.** The meridian mapping experiment contained two types of blocks: horizontal and vertical meridians. The stimuli were wedges of a black-and-white checkerboard (flickering at 1 Hz) radiating out from the fixation spot along the vertical and horizontal meridians, occupying 60° and 30° visual angles respectively. Similar stimuli were used previously to determine the vertical and horizontal meridians that define retinotopic visual areas[17]. Two types of blocks were presented in alternation. Each scan lasted 408 s.

**Stimuli for validation experiments.** After deciding to perform further validation experiments on a specific 25D feature (Figs. 6, 7), the same approach as described above (Selection of representative images for 25D feature space) was followed to select the "positive" and "negative" representative images, by sorting the angles between the 25D

coordinates of all images and the feature of interest. "Positive" blocks were presented in alternation with "negative" blocks. A block of gray-screen preceded each stimulus block and was added at the end of each scan. Each scan lasted 408 s.

## Monkey fMRI experiments

Three male rhesus macaques were trained to maintain fixation on a small spot for juice reward. The fixation spot size was 0.2 degrees in diameter and the fixation window was a circle with a diameter of 2.5 degrees. Monkeys were scanned in a 3 T whole-body scanner (Trio; Siemens Healthcare, Erlangen, Germany) while passively viewing images on a screen. Eye position was continuously monitored at 250 Hz during the scan using an MRI-compatible EyeLink eye tracker system (EyeLink 1000 Infrared Eye Tracker, SR Research, Mississauga, Ontario, Canada) to track pupil position and corneal reflection. Contrast agent (Molday ION, BioPAL, Worcester, Massachusetts, USA) was injected to improve the signal/noise ratio. Whole-brain fMRI data were collected using a gradient-echo echo-planar imaging (EPI) sequence (TR = 2000 ms; TE = 24 ms; flip angle = 80°; slices = 28; matrix = 64 × 64; field of view = 96 mm × 96 mm; 1.5 mm × 1.5 mm in plane resolution; slice thickness = 2 mm; GRAPPA factor = 2). For each session, 5–15 scans were acquired and each scan consisted of 204 functional volumes. Only scans with a total fixation time greater than 80% of the whole scan were included for further analyses. A pair of gradient echo images (echo time: 3.4 and 5.86 ms) with the same orientation and resolution as EPI images were acquired to generate a field map for distortion correction of EPI images. High-resolution T1-weighted anatomical images were acquired using a MPRAGE sequence (TR = 2300 ms; TE = 2.7 ms; inversion time = 1100 ms; flip angle = 9°; acquisition voxel size = 0.5 mm × 0.5 mm × 0.5 mm; 224 horizontal slices). Four or five whole-brain anatomical volumes were acquired and further averaged for better brain segmentation.

## Human fMRI experiments

Four human subjects were scanned for the main stimulus set of 25D object space, and eight subjects participated in the validation experiment (Fig. 7c–e), with three of them engaging in both experiments. All the subjects were scanned with a standard 32-channel phased-array head coil on a Siemens Tim Trio 3.0 T scanner (Erlangen, Germany), while maintaining fixation on a small spot (0.2 degrees in diameter) in the middle of the screen on which the stimuli were displayed. Eye position was continuously monitored at 250 Hz during the scan using an MRI-compatible Eyelink eye tracker system (EyeLink 1000 Infrared Eye Tracker, SR Research, Mississauga, Ontario, Canada). Whole-brain fMRI data were collected using a gradient-echo echo-planar imaging (EPI) sequence (TR = 2000 ms; TE = 30 ms; flip angle = 90°; slices = 50; matrix = 80 × 80; field of view = 240 mm × 240 mm; 3 mm × 3 mm in plane resolution; slice thickness = 3 mm; GRAPPA factor = 1). For each session, 10–12 scans were acquired and each scan consisted of 204 functional volumes. A pair of gradient echo images (echo time: 8 and 10.46 ms) with the same orientation and resolution as EPI images were acquired to generate a field map for distortion correction of EPI images. High-resolution T1-weighted anatomical images were acquired using a MPRAGE sequence (TR = 2300 ms; TE = 3 ms; inversion time = 1000 ms; flip angle = 9°; acquisition voxel size = 0.5 mm × 0.5 mm × 0.5 mm; 176 sagittal slices). Four or five whole-brain anatomical volumes were acquired and further averaged for better brain segmentation.

## Human psychophysical experiments

Nine healthy subjects (four males and five females) with normal or corrected-to-normal vision participated in the psychophysical experiments. All images used in fMRI experiments were presented to the subjects, and they were asked to determine whether the following types of objects were present in each image: animals, mammals, humans, inanimate objects, and artificial objects. The five categories

were not exclusive of each other, therefore multiple labels were allowed for each single image.

## Single-unit recording

Tungsten electrodes (18–20 MΩ at 1 kHz, FHC) were back-loaded into plastic guide tubes. The guide tube length was set to reach approximately 3–5 mm below the dura surface. The electrode was advanced slowly using a manual advancer (Narishige Scientific Instrument, Tokyo, Japan). Neural signals were amplified and extracellular action potentials were isolated using the box method in an on-line spike sorting system (AlphaOmega, Israel). Spikes were sampled at 40 kHz. All spike data were re-sorted using off-line spike sorting clustering algorithms (Spike2, version 7.20, Cambridge Electronic Design Limited, UK). We recorded data from every neuron encountered. Only well-isolated units were considered for further analysis.

**Behavioral task.** Monkeys were head fixed and passively viewed the screen in a dark room. Stimuli for electrophysiology were presented on a LCD monitor (Lenovo LT1913pA). The screen size covered 30.8° × 25.5° visual angles and stimulus size spanned 11°. The fixation spot size was 0.2° in diameter. All images were presented for 150 ms interleaved with 200 ms of a gray screen in random order using custom software. Each image was presented 4–6 times to obtain reliable firing rate statistics. Eye position was monitored using an infrared eye tracking system (ISCAN). Juice reward was delivered every 2–4 s if fixation was properly maintained.

## Data analysis

**fMRI data preprocessing and surface reconstruction.** Analysis of functional volumes was performed using the FreeSurfer Functional Analysis Stream. Volumes were corrected for motion and undistorted based on acquired field map. For localizer experiments and meridian mapping, the resulting data were analyzed using a standard general linear model. For the face contrast, the average of all face blocks was compared to the average of all non-face blocks. For the scene contrast, the scene block was compared to the non-scene blocks. For the stubby-spiky experiment, each of the four block types (stubby animate, spiky animate, stubby inanimate, and spiky inanimate) was compared to three other block types. For meridian mapping, horizontal-meridian block and vertical-meridian block were compared to each other.

Surface reconstruction based on anatomical volumes was performed using FreeSurfer after skull stripping using FSL's Brain Extraction Tool (University of Oxford). After applying these tools, segmentation was further refined manually. For monkey experiments, the result of meridian mapping was projected onto the surface, and the border between V4 and the temporal lobe was delineated. The region of the temporal lobe with significant light responses ($p < 0.001$, paired t-test, two-tailed, control block vs. gray-screen in the 25D object-space experiment, computed using GLM, not corrected for multiple comparisons) was cut out and flattened. To directly compare the functional organization of visual features between subjects, we further registered each subject's brain to a common template of the species to which it belongs, using FreeSurfer's mri_cvs_register function. For monkey and human templates, we used NMT v2[59] and CVS atlas[60], respectively.

**Quantification of feature selectivity to object-space stimuli.** Taking advantage of the periodic nature of the stimulus, Fourier analysis was used to extract the angle and magnitude of each voxel's response to the alternation of positive and negative blocks of each IC. Plotting the magnitudes against the angles of all voxels revealed two clear peaks (Fig. S2a, b). These two peaks represented opposite response polarities, with the left peak indicating a preference for the "positive" block and the right peak indicating a preference for the "negative" block. The angles of 100 voxels with the largest magnitudes in the right peak were then averaged. The difference between the average

angle and the angle of one particular voxel was used to determine the sign of that voxel's response—a difference smaller than π/2 leads to a negative response and vice versa. In order to minimize the effect of daily variations on our results, the response of each voxel was divided by its response to the control block, computed similarly by performing Fourier analysis on its response to the control block and the succeeding block of gray screen. To avoid large variations caused by small denominators, we averaged the absolute values of all significant voxels ($p < 0.001$, paired t-test, two-tailed, control block vs. gray-screen, computed using GLM, not corrected for multiple comparisons) within a distance of 5 voxels to the voxel in the nominator (note that the nominator still represented the response of a single voxel). We used Fourier analysis rather than GLM, because Fourier analysis does not rely on the assumption of the shape of HRF (hemodynamic response function), but we also performed GLM analyses and found that the results were largely consistent (Fig. S2c–k). For GLM, we used the difference between beta values of positive and negative blocks, normalized by that of the control block, to quantify the feature selectivity of each voxel.

After the analysis on functional volumes, the results were projected onto the flat map of the temporal lobe. To enhance the signal quality and better visualize the feature map, we resampled the projections with 2 mm × 2 mm squares, by averaging the 25D responses of all vertices within each square. The 25D vectors were then normalized to unit length. Different square sizes and sampling approaches were also tested, and largely similar results were obtained (Fig. S2c–k).

**The construction of interpretable features and comparison with neural data.** The following features were constructed:

Spatial frequencies:

Using Matlab's fft2 function, the two-dimensional Fourier transform of each image used in the fMRI experiment was obtained. Average magnitudes within seven annuli of the power spectrum, ranging from low to high spatial frequencies, were computed (Fig. S4a). For each frequency band, magnitudes were further averaged across images within "positive" and "negative" blocks of different ICs. The difference between the average magnitudes of two blocks was computed for each IC, and the resulting 25D vector was normalized to unit length.

Color:

For each image, the standard deviation of R, G, and B channels was computed for each pixel and averaged across the whole image to define a "colorfulness index", which was then averaged across images within "positive" and "negative" blocks of different ICs. The difference between the average colorfulness indices of two blocks was computed for each IC, and the resulting 25D vector was normalized to unit length.

PCA axes of AlexNet fc6 layer:

A previous paper characterized a 2D feature map of the monkey ITC, with the two axes corresponding to the top two principal components of the responses of AlexNet units in layer fc6 to a stimulus set of 1224 object images[18]. We performed PCA on the responses of AlexNet units in layer fc6 to a similar set of 1224 images, containing 24 different views of 51 3D objects. Since we could also obtain the 25D coordinates of the 1224 images, we were able to project the top two PCs of layer fc6 into the 25D object space using linear regression. Similar to the previous study, images selected by the two PCs roughly reflected the changes in object shapes from stubby to spiky objects (PC1) and from animate to inanimate objects (PC2; see Fig. S4c, the 2nd row).

Stubby/spiky and animate/inanimate axes:

A different approach was employed to construct the stubby/spiky and animate/inanimate axes. Background-free images of four object types, including: stubby-animate (faces), spiky-animate (animal bodies), stubby-inanimate, and spiky-inanimate objects[18], were presented to AlexNet and the 25D object-space coordinates of those objects were extracted. The difference between the average coordinates of the opposing pairs of objects, i.e., stubby vs. spiky or animate vs. inanimate

object, was computed for each contrast, and the resulting 25D vector was normalized to unit length.

Curvilinear-rectilinear axis:

We collected a set of images composed of multiple curvilinear or rectilinear objects, similar to Fig. 1A of Yue et al.[19]. We presented the images to AlexNet and the 25D object-space coordinates of those images were extracted. The difference between the average coordinates of curvilinear and rectilinear objects was computed, and the resulting 25D vector was normalized to unit length.

Big-small axis:

The same set of stimuli composed of objects with big and small real-world size as in Konkle et al.[15] was presented to AlexNet, and the 25D object-space coordinates of those objects were extracted. The difference between the average coordinates of big and small objects was computed, and the resulting 25D vector was normalized to unit length.

Labels for animals/inanimate objects/mammals/humans/artificial objects (psychophysics):

Using data collected by the aforementioned psychophysical experiment (see Human psychophysical experiments), the number of images containing the five object categories was counted for "positive" and "negative" blocks of each IC separately. The difference between the two numbers was computed for each IC, and the resulting 25D vector was normalized to unit length. The intuition was that, in the ideal case, the average response of a voxel encoding the presence of a particular object type to a block of images would be proportional to the number of images containing that type of object.

Animate/inanimate labels (WordNet):

ImageNet is organized according to the WordNet hierarchy, where each image is labeled with the "category" of an object in the image. We quantified the number of images with animate/inanimate labels in "positive" and "negative" blocks of each IC, and followed the same procedure as for the psychophysical labels to obtain 25D feature vectors.

Comparing neural features with interpretable features:

We used the square of the cosine angle between a neural feature and an interpretable feature to quantify their similarity. Assuming $\vec{v}$ is a vector in an $n$-dimensional space spanned by a set of orthogonal axes: $x_1, x_2, \ldots, x_n$, then it's easy to see that the squares of cosine angles between $\vec{v}$ and $x_i$s add up to 1.

**Feature selection.** To identify the most explanatory features, a feature selection procedure was employed. First, the feature with the largest squared cosine angle in Fig. 3c was selected. We then added one feature at a time to optimize the total explained proportion of neural responses at each step. New features are required to be dissimilar from the ones that have already been selected (cosine angle < 0.5) in order to ensure that they are largely independent of one another. The selected features were orthogonalized to each other using Gram-Schmidt orthogonalization before adding up the squared cosine angles between the visual and the neural features. To determine the optimal number of features, we used a half-split approach: the 25D feature of one location estimated using half the data was fitted with the selected features by projecting the neural features into the subspace spanned by these interpretable features, and then the feature of the other half was compared to the fit (Fig. 3d). The following index was used to quantify the goodness-of-fit (GOF):

$$\text{GOF} = 1 - ||\mathbf{f_2} - \mathbf{f'_1}||^2 \qquad (1)$$

where $\mathbf{f_2}$ is the 25D preferred feature estimated using the 2nd half of the data, $\mathbf{f'_1}$ is the projection of the 1st half's preferred feature ($\mathbf{f_1}$) into the subspace spanned by the selected features ($\mathbf{e_1}$ and $\mathbf{e_2}$ indicate the selected features in Fig. 3d). The index was computed for each location and then averaged across all locations. Note that this

index will be the same as the squared cosine angle when $\mathbf{f_1}$ equals to $\mathbf{f_2}$. We didn't use the squared cosine angle between $\mathbf{f_2}$ and $\mathbf{f'_1}$ directly as GOF, because the polarity of $\mathbf{f'_1}$ should be important for the quality of fitting but would be ignored by the computation of the squared cosine angle. Half of the data used for model training was also used for the feature selection procedure. 500 iterations of half-splitting were performed and the results were averaged. A noise ceiling was also estimated. The preferred feature of the 1st half of the data was directly used to fit that of the 2nd half: $GOF = 1 - ||\mathbf{f_2} - \mathbf{f_1}||^2$. Based on a computational simulation (Fig. S5a), we found that this value would underestimate the true noise ceiling, which is the GOF achieved by the ground truth. Therefore, a correction was made using the estimated relationship between the two (black line in Fig. S5a). Overall, we found seven features performed the best, accounting for 56.3% of the neural features after being normalized by the noise ceiling. Although the exact ordering might differ, the same seven features were identified in 97% of half splits. The actual ordering reported in Fig. 3 is based on the full data. Besides these features, we extracted additional feature dimensions from the neural data, by first orthogonalizing the neural features with respect to the top seven features, and then performing principal component analysis on the orthogonalized features. Since our goal here is to extract dimensions that best align with the preferred features, rather than those that explain the differences between them, both the orthogonalized features and their opposites (-orthogonalized features) were pooled for PCA. Adding two principal components to the seven features performed the best in explaining the neural features, with 82.5% of the noise ceiling. Again, the procedure was performed in the first half of the data and tested on the second half.

**Clustering analysis using a Gaussian mixture model.** We fitted 25D feature map of each monkey with a Gaussian mixture model using the expectation-maximization algorithm (Matlab's fitgmdist function). We constrained the covariance matrix for each component to be diagonal, resulting in 50 parameters per component (25 for the mean, 25 for the variances). We further regularized the covariance matrix by adding a constant ($10^{-5}$) to the diagonal. The number of components (= clusters) was predefined, and the procedure was repeated 500 times for each number of clusters between 2 and 30.

To find the optimal number of clusters, we evaluated the Bayesian information criterion:

$$BIC = -2\log[L] + M\log[N] \qquad (2)$$

where $L$ is the log-likelihood of the model, $N$ is the number of squares and $M$ is the number of parameters in the model, that is, $M = 51C$-1 where $C$ is the number of clusters and the contributions arose from means, variances, and mixture proportions (which have to add to 1). For each number of clusters, BICs were averaged across 500 repetitions of clustering.

**Selecting preferred images of a specific brain region.** The 25D feature selectivity of a specific brain region was first obtained using the methods described above (Quantification of feature selectivity to object-space stimuli). Then we assumed the response of this region to an image is a linear combination of its 25 coordinates. We used $\vec{w}$ to denote the weighting function. 25D coordinates were then averaged for all images in each block and the difference between "positive" and "negative" blocks was computed for each IC. $a_{ij}$ was used to denote the $i^{th}$ coordinate of the difference for the $j^{th}$ IC. Then we have:

$$\vec{R} = \left(a_{ij}\right) \cdot \vec{w} \qquad (3)$$

where $\vec{R}$ is the 25D selectivity measured using fMRI.

Using linear algebra, $\vec{w}$ could be determined by inverting the matrix $\left(a_{ij}\right)$:

$$\vec{w} = \left(a_{ij}\right)^{-1} \cdot \vec{R} \qquad (4)$$

Then we used the same approach as described above (Selection of representative images for 25D feature space) to select representative images for the interested brain region, by sorting the angles between the 25D coordinates of all images and $\vec{w}$. To select the representative images for different subregions resulting from the clustering analysis (Fig. S5j), an additional criterion was added: the angle between the 25D coordinate of the selected image and the average feature of the interested cluster needed to be the smallest of all clusters.

**Image synthesis using a generative model.** Inspired by a recent study on image synthesis[61], we combined a pre-trained generative neural network (BigGAN-deep, see ref. [62]) with AlexNet[38], and used a gradient-based optimization algorithm to synthesize images whose 25 IC coordinates matched that of a target feature (see Fig. S4e). The input to this network consisted of the GAN latent code $\vec{z}$ and a 1000-way class identity, and the output was the 25-dimensional IC coordinate obtained from the responses of fc7 units in AlexNet. The BigGAN network we used was trained to produce images of the size 256 × 256. The latent code $\vec{z}$ was sampled from a truncated normal distribution between [−2, 2], and the class identity was initialized as $\alpha\mathcal{S}(\vec{v})$ where $\alpha = 0.05$, $\mathcal{S}(\cdot)$ is the softmax function, and $\vec{v}$ is randomly sampled from the truncated normal distribution between [0, 1].

The cosine angle $W$ between the 25-dimensional IC coordinate and the target feature can be viewed as a function of the latent code $\vec{z}$ and the vector $\vec{v}$, i.e., $W = f(\vec{z}, \vec{v})$. Since the gradient of $f$ is difficult to calculate, we estimated the gradient using the method of finite difference: For the current vector $\vec{c_0} = (\vec{z_0}, \vec{v_0})$, a set of sample vectors $\vec{c_k}, (k = 1, \cdots, K)$ was derived by adding zero-centered Gaussian perturbations $\vec{p_k}$ with scale $\sigma$ to $\vec{c_0}$: $\vec{c_k} = \vec{c_0} \pm \vec{p_k}$. The gradient of $f$ at $\vec{c_0}$ can be estimated as:

$$\nabla f(\vec{c_0}) = \sum_k (W_{k,+} - W_{k,-})\vec{p_k}/||\vec{p_k}||_2 \qquad (5)$$

where $W_{k,\pm} = f(\vec{c_k})$ and $||\cdot||_2$ denotes the Euclidean norm. Then, we iteratively maximized $W$ using the Adam optimizer method[63].

**Analyzing the validation experiments using GLM.** All the validation experiments in Figs. 6, 7 contained two types of stimulus blocks ("positive" and "negative" blocks), interleaved by gray-screen blocks. T-contrasts between the two stimulus blocks were determined using a standard GLM. In Fig. 7d, beta values of the two stimulus blocks extracted from GLM were baseline-corrected by subtracting the gray-screen block, and were normalized by the maximum value in the region of interest. The normalized responses were averaged across all squares within the region for each scan. The results from multiple scans for the two conditions were compared using a paired t-test (two-tailed).

**Electrophysiology.** Fig. 6e, g, j show peristimulus time histograms of the responses of example neurons to different image conditions, smoothed with a 25 ms sliding window. Average time courses of t-contrasts between the two conditions were computed using the same sliding window (Fig. 6h, k).

In Fig. 6i, the responses to the 25-IC stimulus set were analyzed. In this case, the number of spikes in a time window of 50–400 ms after stimulus onset was counted to measure a neuron's response to each stimulus. For each neuron, the difference between the average responses to positive and negative representative images was

computed for each IC, and the resulting 25D vector was normalized to unit length.

**Comparing feature preferences of human and monkey brains.** To visualize the difference between two primate species, PCA was performed on the 25D preferred features of all individuals pooled together. To balance the contributions from two species, the human data were randomly downsampled to the size of the monkey data. The top two PCs were highly reliable: when we repeated the downsampling procedure 1000 times, the average absolute cosine angle between the PC axes extracted from different repetitions was more than 0.998 for both PCs. A clear difference between the two species was observed on the 2nd PC, with much higher scores in monkeys than in human subjects (Fig. 7a). We found the 2nd PC could be well approximated by linearly combining the "animacy" feature and the "high sf" feature. The average absolute cosine angle between the 2nd PC and the sum of the two aforementioned features is 0.87.

**Using half-split analyses to quantify the dimensionality of neural representation.** For each IC, all scans were randomly split into two groups. 25D feature maps were obtained for both groups using the same procedure described above (Quantification of feature selectivity to object-space stimuli). PCA was carried out on the 25D feature map of one group. Feature maps of both groups were projected onto the PCs, and the correlation between the two projections was computed (Fig. 8b). 500 iterations of half-splitting were repeated, and dimensions with more than 99% positive correlations were considered significant.

To validate our method, we performed the following simulations: PCA was carried out on the original feature map of M1. Top n PCs were used to reconstruct the responses, and a noise term was added (Fig. S7c). A Gaussian white-noise was spatially filtered to match the correlation structure of the noise in real data, which was estimated by the spatial autocorrelation of the difference between two feature maps resulting from random half-splitting as described above (Fig. S7d). We varied the actual dimensionality of the simulation, i.e., the number of PCs used to reconstruct the responses, and the level of noise (as a multiplicative factor, the simulations achieved a similar level of half-split correlations to the neural data at noise level=1 and actual dimensionality=25), and followed the same procedure as we did for the actual data to estimate the number of significant feature dimensions (Fig. S7e, f).

**2D spatial autocorrelation.** 2D spatial autocorrelations were computed for the feature maps (Figs. 8, 9). Feature values belonging to all pairs of squares with the same relative location on the flat map were compared with each other and their correlation was computed. The correlation coefficients at different relative locations constitute the autocorrelation map. With $f(x,y)$ denoting the values of a feature map at location $(x,y)$, the autocorrelation at the relative location of $(dx,dy)$ was estimated as:

$$r(dx, dy) = \frac{n \sum f(x,y) \cdot f(x-dx, y-dy) - \sum f(x,y) \cdot \sum f(x-dx, y-dy)}{\sqrt{n \sum f(x,y)^2 - (\sum f(x,y))^2} \cdot \sqrt{n \sum f(x-dx, y-dy)^2 - (\sum f(x-dx, y-dy))^2}}$$

(6)

where the summation is over all n squares for which a value was estimated for both $f(x,y)$ and $f(x-dx, y-dy)$. In this way, periodic structures of feature maps can be easily identified. One of the most famous applications of spatial autocorrelation in neuroscience is the identification of grid cells in the entorhinal cortex[64].

To better quantify the periodic structure of the autocorrelation map, we outlined a rectangular region (size = 66 mm × 22 mm) with variable orientations around the center of the map, and projected all correlation values within the region onto the long edge. Fourier

transform was then carried out on the projections to extract the magnitudes at different frequencies.

To directly compare the monkey feature map with topographic neural network models (see below: Topographic neural network model), which simulate ITC organization in a single hemisphere, only data from the left ITC was used to compute the autocorrelation map. We fitted the autocorrelation map for each IC with sinusoidal functions at different frequencies and orientations (Fig. 9d, left). Goodness-of-fit, quantified as explained variances, were then averaged across ICs and illustrated as a polar plot (Fig. 9e). The following indices were computed: preferred frequency, frequency tuning width, orientation selectivity, and orthogonality index. Preferred frequency was computed as the explained-variance weighted average of all frequencies (Fig. 9l). Frequency tuning width was defined as the area under the frequency tuning curve in Fig. 9l, with the peak normalized to 1. For orientation selectivity, Fourier transform was first carried out on the orientation tuning curve at a specific frequency as in Fig. 9m. F(1) and F(0) in the inset of Fig. 9m denote the magnitude of the fundamental frequency and the DC component, respectively, and the ratio between the two was used to define the orientation selectivity index (OS). This procedure results in an OS for each frequency. In the middle columns of Fig. 9p–r, the maximum orientation selectivity of all frequencies above 3 cycs/5.8 cm (dashed line in Fig. 9n) is plotted. The orthogonality index was defined as follows: at each frequency of the polar map, its orientation selectivity and preferred orientation were represented by the magnitude and direction of a 2D vector (Fig. 9o). Cross products between all such vectors were computed, and the maximum vector length across all products was used to define the orthogonality index.

Slightly different approaches were employed in the analyses of Figs. 8, 9, because the purposes of the two analyses were not the same: while the main purpose of Fig. 8 was to demonstrate the existence of multiple gradients along different feature dimensions, Fig. 9 aimed to derive a set of scalar indices for quantitative comparisons between animal brains and alternative models. Despite their dissimilarity, the two approaches revealed a similar pattern in monkey temporal lobes: a pair of largely orthogonal feature gradients with different spatial scales.

**Computing geodesic lines on the cortical surface.** To obtain the geodesic line between a pair of vertices on the cortical surface, we modeled the surface as a graph, and then the problem of finding geodesics was transformed into the problem of finding the shortest paths in a graph, with the graph nodes representing vertices and the weights of the edges representing 3D Euclidean distances between neighboring vertices. The implementation includes the following three steps: 1) constructing the edges of the graph; 2) finding the connected graph components; 3) computing the shortest path. For the first step, an edge would be added to the graph between a pair of nodes if their 3D Euclidean distance in the anatomical volume was less than 1 mm and their 2D Euclidean distance on the flat map was less than 1.5 mm. However, this procedure resulted in a large number of connected graph components. Next, we merged pairs of connected graph components with the smallest 1% 3D Euclidean distance between them, which was defined by the minimal distance between all pairs of vertices separately belonging to the two components, by adding edges between the closest pairs of vertices. The 2D flat-map distance between two vertices connected by a new edge was also required to be less than 1.5 mm. Then, the connected graph components were constructed using the classical breadth-first search algorithm. Finally, the shortest paths were solved using the Dijkstra's algorithm.

To identify representative geodesic lines along certain orientations on the flat map as in Fig. S7k, we first placed 2 mm × 2 mm squares on the flat map along two orthogonal orientations (circles in

Figure S7k represent the centers of the squares), then, for each square, the vertex with the smallest 3D distance to the average coordinate of all vertices within the square was selected to represent that square. To simplify the procedure of finding the shortest path, we further required the representative vertex to be part of the largest connected graph component, which contained > 94% of all vertices. Geodesic lines between pairs of representative vertices along the two orthogonal orientations were identified using the methods described in the last paragraph and compared with each other (Fig. S7l).

**Topographic neural network model.** The basic architecture of the convolutional neural networks adopted in our work was introduced by Krizhevsky et al.[38], containing five convolutional layers with max-pooling nonlinearities after layers conv1, conv2, and conv5, followed by three fully-connected layers. Batch normalization technique[65] was applied between the convolutional/fully-connected layer and the ReLU Nonlinearity layer in layers conv3, conv4, fc6 and fc7. We implemented the convolutional neural network model, as well as the self-organizing map in the following section, using the open source machine learning framework PyTorch (https://pytorch.org).

Inspired by a previous study demonstrating the importance of cortical shape on the topographic patterns that could form within it[47], we used the shape of our feature map in monkey and human brains to rearrange the units in the fully connected layer(s) of artificial neural networks (see Fig. 9f and Figure S9g).

Each network was trained using stochastic gradient descent with a batch size of 768 images, a momentum of 0.9 and a weight decay of 0.0005. The learning rates were initialized respectively as 0.01 in the batch normalization layers and 0.03 in the remaining layers, and were decreased by a factor of 10 upon plateau of the total loss throughout 70 epochs of training.

For network models in Fig. 9, the spatial correlation loss function is:

$$L_{\text{Spatial}} = \operatorname*{mean}_{i,j;i\neq j} |C_{ij} - f(d_{ij})|^2 \tag{7}$$

where $C_{ij}$ is the response profile correlation between the model units $i$ and $j$, and $d_{ij}$ is their cortical distance in millimeters. In our implementation, the function $f(d_{ij})$ is given by one of the following formulas (Fig. 9g):

$$f(d_{ij}) = \frac{1}{1 + d_{ij}/s}, s = \frac{1}{2}, 1, \frac{3}{2} \text{ for monkey; } s = 1,2,3 \text{ for human} \tag{8}$$

and

$$f(d_{ij}) = \left(1 - \frac{d_{ij}^2}{2s^2}\right) \exp\left(-\frac{d_{ij}^2}{2s^2}\right) \tag{9}$$
$$s = 2,4,6,8 \text{ for monkey; } s = 4,8,12,16 \text{ for human}$$

The spatial loss and the overall categorization loss (cross entropy loss) were weighted differently, with 10/20/40 times stronger weighting for the spatial loss than the categorization loss:

$$
\begin{aligned}
L &= L_{\text{categorization}} + w\, L_{\text{Spatial}} \\
&= \operatorname*{mean}_{1 \leq b \leq B} \log \frac{\exp(x_{b,y})}{\sum_{n=1}^{N} \exp(x_{b,n})} + w \operatorname*{mean}_{i,j;i\neq j} |C_{ij} - f(d_{ij})|^2, w = 10,20,40.
\end{aligned}
\tag{10}
$$

Here, **x** is the distribution generated by the neural network over the $N$ class labels, $y$ is the target, $N$ is the number of classes, and $B$ is the batch size.

In all panels of Fig. 9 that include population data, results from all networks are presented, except from the right panel of Fig. 9h, where

only networks with the maximum weighting on the spatial loss ( = 40 times the weighting on the categorization loss) are taken into account.

In Fig. 9h, the response correlation $C_{ij}$ of units $i$ and $j$ is given by the Pearson Correlation Coefficient:

$$C_{ij} = \frac{\left(\vec{R}_i - \mu_i\right) \cdot \left(\vec{R}_j - \mu_j\right)}{\|\vec{R}_i - \mu_i\|_2 \|\vec{R}_j - \mu_j\|_2} \tag{11}$$

where $\vec{R}_i$ ($\vec{R}_j$) is the response profile of units $i$ ($j$) to the representative images of the 25D object space used in fMRI experiments (cf. Selection of representative images for 25D feature space), $\mu_i$ ($\mu_j$) is the mean of $\vec{R}_i$ ($\vec{R}_j$), and $\cdot$ denotes the dot product.

Face and object selective units shown in Fig. 9i were defined by computing the following face/object preference metric[30]:

$$d' = \frac{\mu_{face} - \mu_{object}}{\sqrt{\left(\sigma_{face}^2 + \sigma_{object}^2\right)/2}} \tag{12}$$

where $\mu_{face}$ ($\mu_{object}$) is the mean response of a unit to face images (non-face images) used to localize face-selective regions in the temporal lobe[66], and $\sigma_{face}^2$ ($\sigma_{object}^2$) is the variance of the response to face images (non-face images). A unit was defined as face(object)-selective if $d' \geq 0.85$ ($d' \leq -0.85$)[30].

To obtain the 25D feature map for a network model, "positive" and "negative" representative images of the 25D object space were passed through the network. For each unit, its responses to images in the "positive" and "negative" groups of each IC were averaged and compared, resulting in a 25D preferred feature. Since the units were spatially arranged according to the shape of the monkey/human feature map, their locations in relation to the 2 mm × 2 mm squares used for fMRI data analysis were predetermined. The results of all units belonging to the same square were averaged and normalized to unit length. Further analyses of the feature map were the same as for fMRI data.

Our algorithm for training the topographic neural networks is summarized below:

---

**Algorithm** Training the topographic neural network model

---

**Input**: training set $\mathcal{D} = \{(x_n, y_n)\}_{n=1}^N$, validation set $\mathcal{V}$, learning rate $lr$, descending factor
 $\alpha$, and maximum number of epochs $T$.
image transformations (resize, crop and normalize);
build the network and initialize weight $\vec{\mathbf{w}}$ and $\vec{\mathbf{b}}$ randomly;
**repeat**
 reshuffle the samples in the training set;
 **for** $n = 1, \cdots N$ **do**
 select a sample $(x_n, y_n)$ from the training set $\mathcal{D}$;
 Calculate the input and activation values for each layer;
 obtain the weighted sum of spatial loss and categorization loss by Eq. (10);
 back propagation and update the weight $\vec{\mathbf{w}}$ and $\vec{\mathbf{b}}$;
 **end**
 $t \leftarrow t + 1$;
 **if** the total loss on the validation set has stopped decrease, **then**
 $lr \leftarrow \alpha \times lr$;
**until** $t = T$;
**Output**: well-trained neural network

---

**Self-organizing map.** Inspired by a previous work[31], we built a self-organizing map[67] using the 25D IC coordinates as inputs to mimic the topographic organization of visual features. There were 6400 units in the output layer of SOM, arranged according to the shape of the monkey feature map (see Fig. S9a).

In each step of training, the winner unit $u$ in the output layer was defined as the unit whose weight vector was closest to the input feature:

$$u = \arg\min_{v} ||\vec{\xi} - \vec{w}_v||_2 \tag{13}$$

where $\vec{\xi}$ is the input feature, and $\vec{w}_v$ is the afferent input weight from the input feature to the unit $v$. The afferent input weight $\vec{w}_v$ was then updated using the following formula:

$$\vec{w}_v = \frac{\vec{w}_v + \eta_v\left(\vec{\xi} - \vec{w}_v\right)}{||\vec{w}_v + \eta_v\left(\vec{\xi} - \vec{w}_v\right)||_2} \tag{14}$$

with the learning rate $\eta_v = \eta f(d_{uv})$ for the unit $v$. Here, $\eta$ is the learning rate for the unit $u$, the function $f(\cdot)$ is given either by Equations (8), (9), or a gaussian function with a spatial scale comparable to the prior study[31]—the scale parameter $\sigma$ of the gaussian function was set to be 1.25 mm, so that the function value was 10% of its peak at 2.7 mm away from the center, matching the experimentally measured cortical point-spread function in the macaque visual cortex[68].

The SOM was iteratively trained for $T = 10^7$ steps on the validation set of the ImageNet dataset. The afferent input weights were initialized randomly. In every training step $t$, the learning rate was given by $\eta^t = T/(T + 2t)$. At the end of the training, the corresponding 25-dimensional feature map was obtained in the same manner as for the convolutional neural networks.

Our algorithm for training the self-organizing maps is summarized below:

---

**Algorithm** Training the self-organizing map (SOM)

**Input**: training set (input feature) $\mathcal{D} = \left\{\vec{\xi}_n\right\}_{n=1}^N$, number of units in input and output layers $N_i$
and $N_o$, and scale parameter $\sigma$, training steps $T$.

initialize weight $\vec{w}_v$ randomly and $t \leftarrow 0$;

**repeat**

 reshuffle the samples in the training set;

 **for** $n = 1, \cdots N$ **do**

 select input feature $\vec{\xi}$ from the training set $\mathcal{D}$;

 obtain the winner unit $u$ using Eq. (13);

 update the weight $\vec{w}_v$ using Eq. (14);

 $t \leftarrow t + 1$;

 **end**

**until** $t = T$;

**Output**: $\vec{w}_v$

---

## Reporting summary

Further information on research design is available in the Nature Portfolio Reporting Summary linked to this article.

## Data availability

Natural images used in the main stimulus set are available in the ImageNet (https://image-net.org/download.php). Face images used in the four-object-type stimulus set are available in the FEI database (https://fei.edu.br/~cet/facedatabase.html). The raw data supporting the current study are available under restricted access because of the size of the data and the complexity of its structure; access can be obtained by contacting Le Chang (lechang@ion.ac.cn). Source data are provided with this paper.

## Code availability

Custom codes for neural network training and related analysis have been deposited at Zenodo and are publicly available at the time of publication (https://doi.org/10.5281/zenodo.8053796).

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

## Acknowledgements

This work was supported by STI2030-Major Projects (2022ZD0204600), Natural Science Foundation of China (31822023 and 31871053), Strategic Priority Research Program of CAS (XDB32000000), and Shanghai Municipal Science and Technology Major Project (2018SHZDZX05) to L.C. We thank the 3 T core facility (CEBSIT) for technical support, the Macaque Animal Facility of CEBSIT for animal care, Ling Long for help with psychophysical experiments, Gepei Zhang for stimulus preparation, and Dr. Shuo Wang and members of the Chang lab for helpful comments on the manuscript.

## Author contributions

M.Yao, B.W., M.Yang, H.J., C.F., and Y.C. performed the experiments; M.Yao, B.W., M.Yang, J.G., and L.C. analyzed the data; J.G. constructed the neural network models; L.C. conceived the project; H.H. and L.C. supervised the project; all authors wrote the paper together.

## Competing interests

The authors declare no competing interests.
