## [Peer Review File · Nature Communications]

High-dimensional topographic organization of visual features in the primate temporal lobeReviewer #1 (Remarks to the Author):

Yao and colleagues performed an impressive series of work to investigate a key question in visual cognition: how was the overall functional architecture in temporal cortex. The authors constructed a 25-dimension feature space and selected 50 representative images for each of the 25 dimensions. Using fMRI, they imaged cortical responses from monkeys and humans to these images. They were then able to represent full characteristics of the IT voxels in this 25D space, and established relationship with a set of main visual features. The authors found topographic organization for the representative visual features along the temporal cortex, including new features (e.g. residual features) that have not been characterized before. They further verified their new findings with electrode recordings guided by fMRI mapping. This study took a novel approach to study an interesting yet difficult question in the visual neuroscience field. The results are highly relevant to understand the overall organization of the temporal cortex. The paper is well written and technically sound. In general, this is a high-quality research work that is acceptable by Nature Communication. I only have several suggestions for authors to consider in improving the manuscript.

Major:

The authors selected 25 ICs based on ICA analysis on kernels of AlexNet FC7. This is a clever approach. However, this is based on a hypothesis that AlexNet FC7 is similar to primate IT cortex, which is still under intensive debates. The authors should discuss the relevant literature as well as how much it will affect their conclusions. In addition, the 25D feature space is a result of training with two-dimensional natural images, which may not fully represent the actual visual world. Some key visual information is not represented in the ImageNet database. For example, binocular disparity and temporal dynamic information. I suggest the authors add some discussion on this issue.

This modeled cortex includes the full temporal lobe, and is not limited to IT. The main body of results also did not differentiate IT from the dorsal part of temporal lobe, which mainly involves in processing of visual motion and auditory information. The authors mentioned: "... because we observed ordered and continuous feature selectivity over the entire temporal lobe." However, the maps in Figure 2A were not tested with statistical methods. It is not known whether those dorsal areas were significantly responding, and how they different from IT. I suggest the authors discuss this concern.

The manuscript can provide more comparative discussion on their results and earlier findings on functional architectures in IT. For example, color patch is one of the most prominent features in IT. Many researchers including the corresponding author of this manuscript have studied color patches in macaques. The color feature, however, was not among the top features detected in the present findings. The authors may want to discuss the reasons.

IT cortex is traditionally divided into 3 sub-areas. Many functional features are also divided into 3 parts (e.g. face and color). The topographic map in the present study did show certain periodicity in some features but not for others. The authors may want to commend on the relationship between these two.

Minor:

The above-chance correlation shown in Figure 2E needs some explanations. Is this result used to justify the choice of the 25 ICs, or just because the 25 ICs were not independent (mentioned in line 43 page 16)?

Page 8 line 18: Why examined 6 features instead of 7 or 9 that were described earlier.

Page 11 line 4: (Figures S4E and 4G) should be

Reviewer #2 (Remarks to the Author):

In this paper, Yao et al leverage DNNs to delineate a 25D visual feature space based on the responses of DNN units to a large set of natural images, which they then used in humans and monkey fMRI experiments to map out the functional organization of the temporal lobe. With this clever and novel approach, the authors move beyond semantic labels and categories to understand the general principles underlying the functional organization of the temporal lobe, including the inferotemporal cortex, a highly studied region involved in object recognition. This approach also allowed the authors to investigate the feature selectivity of previously unexplored parts of the temporal lobe, which they corroborated with both fMRI and electrophysiological experiments. This is an important paper implementing a bias-free method to parcellate the temporal cortex, using rigorous and thorough experiments and an impressive amount of data.

My biggest concern is the implication that this gradient of multiple selectivities that the brain seems to share with DNNs is that it must reveal something about natural image statistics, rather than something about the brain. These must be parameters that images vary mostly along, rather than some optimal coding scheme.

Having said that, I found this to be a dense paper where I felt I was missing some details/simplified explanations of the methods, making it difficult for me to follow the paper. I mention a few specific comments about this below –

1. How separable were the 25 independent components that were extracted based on the responses of the DNN units? Is it possible that some of them overlapped?
2. Connected to the previous question, how do I interpret the “positive” and “negative” images for each IC? Do the two groups represent two extremes on an axis (for instance, hypothetically if an IC axis was color, would an image in the positive group contain color and an image in the negative group contain less color, as is also evident in the representative images shown in Fig. S3B2?). Further, if the representative images were extracted using the smallest and largest angle between their IC coordinates and the IC axis, was it possible that some of the images overlapped with other ICs (for example, in the schematic illustration of the IC axis and angle in Figure 1B, the largest angle to IC1 could be close to the smallest angle of IC3).
3. It says in the methods that 50 representative images were selected for each positive and negative group of each IC, and each image was shown for 500ms, but each block was only 24s long? Does this mean only 48 images were shown? Also, did the positive and negative blocks in each scan contain the exact same images which were repeated? That is, the same block of positive images was shown 8 times and the same block of negative images was shown 6 times in one scan (in the example stimulus sequence in Fig. 1E). If so, was the order of the images within a block randomized each time?
4. It could be useful if the authors indicated exactly where on the surface (Fig. 1F1) corresponds to the flat map (Fig. 1F2).
5. In Fig. 2, what do the colors in the IC maps shown in A correspond to? Does close to 1 indicate the response to positive images and close to -1 indicate the response to negative images? This should be made clearer in the legend, as well as the text.
6. I had some difficulty understanding what exactly “visual features” refer to, and how they were compared with “neural features” in the section “explaining the feature preference map using interpretable visual features”. Do the “visual features” simply refer to the activations of the DNN units, while the “neural features” refer to the neural responses obtained from the fMRI experiments? If not, how exactly were the “visual features” measured/extracted?
7. If this is possible, it could be useful to project the feature map shown in Fig. 4A or Fig. 5A1 on the flat map of the brain with the sulci visible underneath the feature map – this could be helpful in anatomically orienting the reader and understand the location of the feature maps in relation to the different regions in the temporal lobe.
8. In the section “testing new predictions based on the 25D preference maps”, the authors say that positive and negative representative images were selected for the 1st residual feature (or feature 8), which were then used in an fMRI experiment. I assume that this means that the representative images of the 1st residual feature used in this fMRI experiment were selected from the 200k natural images from ImageNet. Were the synthesized images (shown in Fig. S3D3) not used in any experiments?

Reviewer #3 (Remarks to the Author):

High-Dimensional topographic organization of visual features in the primate temporal lobe

The authors collected monkey and human fMRI data in response to images based on the independent components of the representational spaces of late stage Alexnet Model. They conduct numerous complex analyses, and argue their approach “fills-in gaps” of high-dimensional feature maps of primate temporal lobe, and “pinpoints critical differences” in the topographic maps of monkeys, humans, and “AI”.

I found the approach to measure responses in both monkey and humans and explore convergences in their topographies to be a novel contribution. However, I found the manuscript to be not well-situated in the current literature, with overly broad scope, overly complex analyses, and unclear contributions to current understanding of the ventral visual stream topography in primates. I detail these below.

1. The work is not situated in the literature well. The work makes claims that are already known, and sets up the problem not recognizing both earlier, and more recent progress.

For example, the motivating set up is: “a panoramic view of topographic organization is still lacking”. This is false. Consider work by Haxby et al., 2011; Huth & Gallant; Konkle & Caramazza 2017; Conway’s 2018 Ann Review; not to mention Bao et al., 2020! There is extensive work charting the large-scale organization of object information and tuning, both empirically and theoretically. As a specific example, consider the title: “high dimensional organization of visual features in primate temporal lobe”. Note Haxby et al., title from 2011 (a decade ago!): “A Common, High-Dimensional Model of the Representational Space in Human Ventral Temporal Cortex.” And Bao et al 2020: “A map of object space in primate inferotemporal cortex”. There are also proposals for a panoramic view of the topographic organization: Konkle & Carmazza, 2013: “Tripartite Organization of the Ventral Stream by Animacy and Object Size.” Even earlier work by Levy, Malack, and Hasson (e.g. Hasson et al., 2004) charted a large-scale map of object cortex (with no gaps!) and presented a theory for its large-scale formation.

Additionally, there are now a number of models that take deep net spaces and map them topographically in different ways (e.g. Blauch et al., Keller & Welling, Doshi & Konkle, Zhang et al.,). These ‘near-neighbor’ papers are also relevant to the claims and theoretical positioning of this work.

2. The framing is overly ambitious, (as are the scope of analyses included in the paper, see below). This makes the contributions of the paper difficult to assess and feel unsubstantiated (undermining the tremendous effort and impressive computational work that went into this project!). This comment related to that above, and can be further evidenced by claims like that of the final sentence of the abstract (“pinpointing the critical differences in the functional organization of high-level visual areas between monkey, human, and artificial intelligence”).

3. The method of collecting and analyzing the data had some assumptions that gave me pause and which I needed more explanation for their implications on subsequent analyses and results.

3.a. Data analysis: one early choice you made was to look at activations along ICs relative to a control block of 50 images. This confused me, as each one of these 50 images has some value on each IC. By chance you may have be partially controlling away ICs based on this “control” block. Clarifying the logic and implications of this choice would be important—do you really need to do this step?

3.b. Data analysis: The images were selected based on the 25 ICs to reflect the positive and negative ends, presented in a blocked design. This means that by definition you will be measuring how these pre-defined axes project into the (stable!) tuning of the object cortex. Of course there will be a ‘topography’ of these ICs, because each voxel/vertex has some tuning function, which bears some relationship to the ICs, and so you can just make a map of ICs. This is assumed by the experimental design and required by the data analysis pipeline and not testing anything per se. The similarity matrix of the ICs should be similar because there’s stable tuning of the visual

system... the degree to which they are dissimilar really means you have noisy data, I think? And the claim that you have multiple oriented gradients... I think this is required from your analysis rather than a discovery where alternate patterns of results were possible. Is there any way you could have found something else? I'm not sure and it was hard to follow complex analysis procedures used here.

3.c. You also talk about the dimensions of ICs, but only sampled positive/negative poles, so it is unclear whether you have any evidence for dimensional gradients.

4. The analysis pipeline was too complex and hard to follow

4.a. You went from 25 ICs (Figure 2), to a map of 21 'intuitive features' (Figure 3c) to a sub-selection of 9 features (Figure 3F), to a set of 3 (Figure 4a). At each step the transformations and assumptions, for me, further obscured insights into the tuning, rather than clarify them. Staying closer to the data, revealing the structure in the maps, and distilling the key insights would clarify the contributions

4.b. Relatedly, the use of low-, mid-, and high-level features and the logic around how you ruled out mid-level features was faulty. This was not an experimental study designed to tease these levels of representation apart, and I think they confuse the issue. Your study was aimed to see if the ICs of the late stage of an Alexnet give insight into the large-scale organization of monkey and human object cortex. Focus on that. Delineating what 'level' these features are at requires different data (e.g. see Long et al., 2018, Jagadeesh & Gardner, 2022), and is itself a rather thorny issue that I'd recommend not focusing on here.

5. I found the scope of included results was too extensive to vet everything. For example, by the end of the results section, you implemented your own TDANNs to see if "orientation-dependent organization ... naturally emerged in networks trained with orientation-independent spatial loss"--this felt like an unnecessarily large addition, as there are again many many choices here, and it is very hard follow the logic of which choices you made, and how you assessed it, and how your supported your claims. This could be a whole additional paper! Further, this is not the only topographic deep net model and so claims that these deep nets don't replicate broader orthogonal gradients are, e.g., not in line with approaches like Zhang et al, and Doshi & Konkle, 2022

6. Take home results unclear. By the end of the results section, I was not left with a clear sense of the key take home points of how my understanding of the object-cortical organization should be updated.

7. Discussion did not appropriately acknowledge prior work. As noted in above, claiming there is 'no general principle underlying the topographic organization of primate temporal lobe' fails to acknowledge prior work (c.f. Bao, Konkle, Hasson, Conway, Haxby, Huth). I found the discussion around the level of representation of tuning to be logically unsound.

Despite these critics, I feel there are real areas of novelty that could be distilled and treated with a more careful curated scope and lead to an important, high-impact contribution. For example, this is, to my knowledge, once of the very few condition-rich datasets collected in both humans and monkeys, and I don't know of any careful topographic work comparing their spatial organizations. Further, you highlight that your data reveal fine-grained differences in how humans and monkeys represent in animate objects, and in their topography. Clarifying these empirical results, simply and clearly, would be an important and--to my knowledge--novel contribution.

Further, clarifying that you're using the ICs of an Alexnet as a lens into the tuning is going to be important in how you scope your claims. For example, it is important to acknowledge and discuss the degree to which your claims/results depend on their being a good alignment between these features spaces, and what happens if these ICs are not in good alignment with the true underlying tuning functions of the ventral visual stream voxels/vertices.

Overall, it is clear that the authors have carried out an impressive amount of work, both in the dataset collection and in the analytical approaches. There is likely more than one paper here. However, as currently written, scoped, and situated, the novel contributions of the work are

obscured and unsubstantiated.

We thank all three reviewers for their careful reading of our paper and their excellent suggestions. By addressing the concerns raised and incorporating our responses into the manuscript, we believe the paper has been significantly improved. Below, we will address each of the specific points raised by the reviewers in detail, with the original comments in blue, our responses in black, and the text added to the manuscript underlined.

Reviewer #1:

Yao and colleagues performed an impressive series of work to investigate a key question in visual cognition: how was the overall functional architecture in temporal cortex. The authors constructed a 25-dimension feature space and selected 50 representative images for each of the 25 dimensions. Using fMRI, they imaged cortical responses from monkeys and humans to these images. They were then able to represent full characteristics of the IT voxels in this 25D space, and established relationship with a set of main visual features. The authors found topographic organization for the representative visual features along the temporal cortex, including new features (e.g. residual features) that have not been characterized before. They further verified their new findings with electrode recordings guided by fMRI mapping. This study took a novel approach to study an interesting yet difficult question in the visual neuroscience field. The results are highly relevant to understand the overall organization of the temporal cortex. The paper is well written and technically sound. In general, this is a high-quality research work that is acceptable by Nature Communication. I only have several suggestions for authors to consider in improving the manuscript.

We thank the reviewer for appreciating our study and the helpful comments.

Major:

The authors selected 25 ICs based on ICA analysis on kernels of AlexNet FC7. This is a clever approach. However, this is based on a hypothesis that AlexNet FC7 is similar to primate IT cortex, which is still under intensive debates. The authors should discuss the relevant literature as well as how much it will affect their conclusions. In addition, the 25D feature space is a result of training with two-dimensional natural images, which may not fully represent the actual visual world. Some key visual information is not represented in the ImageNet database. For example, binocular disparity and temporal dynamic information. I suggest the authors add some discussion on this issue.

We thank the reviewer for pointing this out. The choice of the network and the corresponding layer was supported by previous studies showing that units in this layer could well explain the responses of IT neurons to images of objects and faces (Cadieu et al., 2014; Chang et al., 2021), with the percentage of variance explained >50%. Admittedly, the 25 IC dimensions would not fully explain the neural responses to natural images, even the static ones. We think what we have done is like projecting the true neural tuning in the space spanned by all visual features into a subspace spanned by the IC dimensions (Figure R1, from the solid arrows to the dashed arrows). Classical localizers have typically used a pair of conditions to compute a contrast, which can be

thought of as a projection into a 1D subspace (e.g., face localizer = faces vs. objects). By using a high-dimensional space, we think that we can get closer to the true neural tuning and thus better capture the relationship between the tuning of different subregions. In the future, if additional features not included in our 25 ICs are tested in the primate temporal lobe, this new knowledge could be integrated with our results to provide a more comprehensive map (going from the dashed arrows back to the solid arrows). This should also apply to visual information not represented in the ImageNet images, including binocular disparity and temporal dynamics information.

We have added Figure R1 as Figure S1E and the following in the discussion section (pages 22-23): “By analyzing the neural tuning using features derived from AlexNet, what we have done is like projecting the true neural tuning into a subspace spanned by AlexNet features (Figure S1E). Although AlexNet has been shown to be an excellent model of high-level visual areas (Cadieu et al., 2014; Bao et al., 2020; Chang et al., 2021), there is still likely to be a significant fraction of neural responses that cannot be explained by these features. Future work is needed to explore additional features to approach the true tuning functions. In addition to the features of static 2D images, visual information not represented in the ImageNet images, such as binocular disparity and temporal dynamics information, should also be considered.”

Figure R1. In this diagram, we assume that the selectivity of each neuron can be represented by an axis in the space spanned by multiple features (the “axis” coding scheme, Chang and Tsao, 2017; Bao et al., 2020), which is indicated by a solid colored arrow. Dashed arrows indicate the projections of these axes (solid arrows) into the subspace spanned by the IC dimensions. An additional feature orthogonal to the ICs is indicated by the z-axis.

This modeled cortex includes the full temporal lobe, and is not limited to IT. The main body of results also did not differentiate IT from the dorsal part of temporal lobe, which mainly involves in processing of visual motion and auditory information. The authors mentioned: “... because we observed ordered and continuous feature selectivity over the entire temporal lobe.” However, the maps in Figure 2A were not tested with statistical methods. It is not known whether those dorsal areas were significantly responding, and how they different from IT. I suggest the authors discuss this concern.

To test the significance of neural selectivity, we used a half-split approach to quantify the reliability of the 25D preferred feature across the temporal lobe, and found that most locations, including those in the dorsal part, show consistent preferences (Figure R2A). Subregions

specialized in processing visual motion (MT and MST) and auditory information were identified based on a macaque atlas (CHRAM, Jung et al., 2021). The preferred features of a specific subregion in each hemisphere were averaged (see Figure R2B for the six average features of the auditory area). The average similarity between the average preferred features of all six hemispheres was 0.82 for auditory areas and 0.72 for motion areas ($p < 0.01$ in both cases, tested by randomly shuffling the 25 IC dimensions for 2000 times), suggesting consistent tuning across hemispheres. We then compared the preferred features of different areas. An SVM was trained to discriminate preferred features of dorsal areas from those of IT, resulting in reasonably good classifications (accuracy=87% between auditory areas and IT, =80% between motion areas and IT; the same number of features were randomly sampled for the two areas being compared for 1000 times, so the chance level=50%; for the distribution of the distance of individual feature to the optimal hyperplane, see Figure R2C). Finally, the representative images of the differential axis between dorsal areas and IT were selected to visualize the difference in their feature preference. While the negative end contains mostly of isolated objects (IT), the images of the positive end are more crowded (dorsal areas) (Figure R2D).

We have now added the analyses as Figure S3 and the following text on page 5:

It's also important to note that our feature maps include not only the ITC, but also the dorsal part of the temporal lobe, which is mainly involved in processing visual motion and auditory information. Further analyses suggest that the estimated preferred features of the dorsal areas are reliable and distinct from those of the ITC. (Figure S3).

Figure R2. Neural tuning in the dorsal areas of the temporal lobe.

A) Reliability of neural tuning across the temporal lobe. All scans were randomly split into two halves and the similarity at each location between the 25-D preferred features of two halves was

quantified using their cosine angle. We considered the tuning of a location to be significant if 95% of 500 iterations of random half-splitting resulted in a positive cosine angle. Dashed lines indicate the fundus of STS. B) Based on the CHARM atlas (Jung et al., 2021), preferred features of all squares within the auditory area of each hemisphere were averaged, resulting in six 25D preferred features, which are shown here. C₁) SVM models were trained to discriminate the preferred features of the auditory areas and IT (TE+TEO) of all three monkeys. Since there were fewer auditory features than IT features, the same number of features as the auditory features were randomly sampled from IT to train the classifier. The distribution of the distance to the optimal hyperplane for one iteration of random sampling is plotted. The average accuracy for 1000 iterations of random sampling is 87%. C₂) Same as (C₁), but between the motion areas (MT+MST) and IT. The average classification accuracy is 80%. D₁) The difference between the average preferred feature of the auditory areas and that of IT was computed. The positive and negative representative images of the differential feature are shown. D₂) Same as (D₁), but between motion areas and IT.

The manuscript can provide more comparative discussion on their results and earlier findings on functional architectures in IT. For example, color patch is one of the most prominent features in IT. Many researchers including the corresponding author of this manuscript have studied color patches in macaques. The color feature, however, was not among the top features detected in the present findings. The authors may want to discuss the reasons.

We think that the main reason why we didn't observe a clear color-related selectivity is probably the stimulus set we used. Previous studies have typically compared color images with grayscale versions of the same images, such as gratings, to localize color areas (Lafer-sousa and Conway, 2013; Chang et al., 2017). In addition, these studies suggest that the color areas are also tuned to non-color dimensions, such as shape or category. Compared to the conventional color localizer, our stimulus set consisted mostly of natural images, which are much richer in the object shape and category (as these images were collected for the purpose of testing object categorization ability), but may not cover a similar range of color variations (as color is intentionally manipulated in the conventional color localizer). Therefore, the selectivity for color dimensions may be dwarfed by that for non-color dimensions in our experiment in comparison to previous color studies. We have now added the following on page 23:

While previous studies have identified several color selective regions in ITC, we didn't see a strong tuning to color in our dataset. We think it's likely due to the difference in the stimulus sets used. Previous studies have typically compared color images with grayscale versions of the same images, such as gratings, to localize color areas (Lafer-sousa and Conway, 2013; Chang et al., 2017). Compared to the conventional color localizer, our stimulus set consisted mostly of natural images, which are much richer in the object shape and category (as these images were collected for the purpose of testing object categorization ability), but may not cover a similar range of color variations (as color is intentionally manipulated in the conventional localizer), favoring tuning to non-color dimensions in our experiment.

IT cortex is traditionally divided into 3 sub-areas. Many functional features are also divided into 3 parts (e.g. face and color). The topographic map in the present study did show certain periodicity

in some features but not for others. The authors may want to comment on the relationship between these two.

We thank the reviewer for pointing this out. Indeed, while some of the features show a clear periodicity along the posterior-anterior axis (such as IC1 in Figure 2A), consistent with the division of IT into three subregions, others do not (such as IC17 in Figure 2A). We think the main reason is the richness of our stimulus set, e.g., IC17 contains isolated objects in the positive block and crowded scenes in the negative block (Figure R3), which is not typical for conventional functional localizers: for the face localizer, previous research compared responses to isolated faces and isolated objects; for the color localizer, responses to colored and grayscale gratings were compared. One localizer stimulus that may be considered similar to features like IC17 is the place localizer (Kornblith et al., 2013), which compares the response to images of scenes with that to isolated objects. Indeed, the localized place areas are organized in a way that is less consistent with the 3-part scheme and contain fewer subregions (1 or 2) than face/color patches.

We have now added Figure R3 as Figure S1F and the following text (page 23):

Functional subdomains in ITC have been shown to consist of three parts (Bao et al., 2020), consistent with the periodic pattern along the posterior-anterior axis in some dimensions of our feature map (such as IC1 in Figure 2A). However, this pattern was not observed for other dimensions (such as IC17). We think this is due to the richness of the stimulus set we used, e.g., IC17 contains isolated objects in the positive block and crowded scenes in the negative block (Figure S1F), which is not typical for conventional functional localizers.

Figure R3. Representative images for IC17.

Minor:

The above-chance correlation shown in Figure 2E needs some explanations. Is this result used to justify the choice of the 25 ICs, or just because the 25 ICs were not independent (mentioned in line 43 page 16)?

Based on our procedure, the 25 ICs are independent from each other at the image level (correlation between the projections of 20k ImageNet images onto two different ICs=0), but they are not independent at the level of neural responses—the temporal lobe may represent two IC features similarly. The detailed correlation structure in a brain region (e.g., matrices in Figure 2B) captures the way how visual information is represented, as in the classical representational similarity analysis. If the feature preference maps of two IC dimensions are very similar for a brain area (correlation close to 1 between the two ICs), this brain area would not distinguish between

these two dimensions. For example, if IC1 and IC2 represent the presence of mammals and reptiles, respectively, a high similarity between the maps of the two ICs suggests that the brain area encodes the presence of an animal in general, but not a specific type of animal (mammal or reptile). Therefore, the above-chance correlation between these correlation matrices in Figure 2E indicates that the neural representation is consistent across hemispheres and subjects.

We have now added (page 5):

A similarity matrix was then computed by correlating the maps of all pairs of ICs (Figure 2B). As in the classical representational similarity analysis, this matrix captures the way how visual information is represented. By computing the correlations between the similarity matrices, we found that visual features were represented similarly across hemispheres, individuals, and even species (Figures 2C-E).

Page 8 line 18: Why examined 6 features instead of 7 or 9 that were described earlier.

This is a typo. It should read "9 features", six of which are shown in Figure 3E and three of which are shown in Figure S5D₁, and has now been corrected.

Page 11 line 4: (Figures S4E and 4G) should be (Figures S4E and G).

We have changed it to "Figures S5E and G" (the figure is currently Figure S5).

References:

- Cadieu, C. F. *et al.* Deep neural networks rival the representation of primate IT cortex for core visual object recognition. *PLoS Comput Biol* **10**, e1003963, doi:10.1371/journal.pcbi.1003963 (2014).
- Chang, L., Egger, B., Vetter, T. & Tsao, D. Y. Explaining face representation in the primate brain using different computational models. *Curr Biol* **31**, 2785-2795 e2784, doi:10.1016/j.cub.2021.04.014 (2021).
- Bao, P., She, L., McGill, M. & Tsao, D. Y. A map of object space in primate inferotemporal cortex. *Nature* **583**, 103-108, doi:10.1038/s41586-020-2350-5 (2020).
- Chang, L. & Tsao, D. Y. The Code for Facial Identity in the Primate Brain. *Cell* **169**, 1013-1028 e1014, doi:10.1016/j.cell.2017.05.011 (2017).
- Jung, B. *et al.* A comprehensive macaque fMRI pipeline and hierarchical atlas. *Neuroimage* **235**, 117997, doi:10.1016/j.neuroimage.2021.117997 (2021).
- Lafer-Sousa, R. & Conway, B. R. Parallel, multi-stage processing of colors, faces and shapes in macaque inferior temporal cortex. *Nat Neurosci* **16**, 1870-1878, doi:10.1038/nn.3555 (2013).
- Chang, L., Bao, P. & Tsao, D. Y. The representation of colored objects in macaque color patches. *Nat Commun* **8**, 2064, doi:10.1038/s41467-017-01912-7 (2017).
- Kornblith, S., Cheng, X., Ohayon, S. & Tsao, D. Y. A network for scene processing in the macaque temporal lobe. *Neuron* **79**, 766-781, doi:10.1016/j.neuron.2013.06.015 (2013).

Reviewer #2:

In this paper, Yao et al leverage DNNs to delineate a 25D visual feature space based on the responses of DNN units to a large set of natural images, which they then used in humans and monkey fMRI experiments to map out the functional organization of the temporal lobe. With this clever and novel approach, the authors move beyond semantic labels and categories to understand the general principles underlying the functional organization of the temporal lobe, including the inferotemporal cortex, a highly studied region involved in object recognition. This approach also allowed the authors to investigate the feature selectivity of previously unexplored parts of the temporal lobe, which they corroborated with both fMRI and electrophysiological experiments. This is an important paper implementing a bias-free method to parcellate the temporal cortex, using rigorous and thorough experiments and an impressive amount of data.

My biggest concern is the implication that this gradient of multiple selectivities that the brain seems to share with DNNs is that it must reveal something about natural image statistics, rather than something about the brain. These must be parameters that images vary mostly along, rather than some optimal coding scheme.

We thank the reviewer for the positive assessments and the constructive comments. We agree with the reviewer that the features extracted from DNNs are some kind of image statistics. However, we would like to mention that recent studies have shown that these DNN features accurately predict the responses of high-level visual neurons (Cadieu et al., 2014; Schrimpf et al., 2018; Jozwik et al., 2019), even better than the semantic categories (Bao et al., 2020), which were previously thought to be encoded by IT neurons. Traditionally, fMRI studies measured how voxel responses were tuned to a few predefined features, such as color and category. This type of study may identify a brain region that is selective for a particular feature, but it's hard to conclude that this feature is exactly what this brain region is encoding. Our understanding of the tuning and selectivity of visual areas has been continually updated by introducing new features into the experiments, gradually approaching the "true" tuning of our brains. In this context, what we did in this study can be seen as an extension of the classic work using a set of features that have been shown to be most efficient in explaining neural responses. It's also important to note that we used multiple features simultaneously. Characterizing the neural tuning using multiple features has two main advantages: 1) Compared to a single feature, we get closer to the true neural tuning by examining more aspects of visual information. 2) Within a high-dimensional space of DNN features, it is possible to compare the similarity between the neural tuning of different brain regions in this high-dimensional space, which is crucial for revealing the topographic organization of the visual cortex (e.g., see the parcellations in Figure 3F and Figure 4). Overall, while these features indeed capture mostly natural image statistics, the recent evidence that these features accurately predict the neural responses and our decision to use a large number of features simultaneously allow us to reveal the detailed functional organization of the primate temporal lobe.

We have now added (page 22):

Although the features we used are extracted from deep neural networks (DNN) but not from neural responses, the recent evidence that DNN features accurately predict the responses of

visual neurons (Cadieu et al., 2014; Schrimpf et al., 2018; Jozwik et al., 2019) and our decision to use a large number of features at the same time allowed us to 1) obtain a reasonably good approximation of the true neural tuning; 2) reveal the topographic organization of the primate temporal lobe in detail.

Having said that, I found this to be a dense paper where I felt I was missing some details/simplified explanations of the methods, making it difficult for me to follow the paper. I mention a few specific comments about this below –

1. How separable were the 25 independent components that were extracted based on the responses of the DNN units? Is it possible that some of them overlapped?

Since the independent components (ICs) can be understood as estimated independent sources of the original signal and every image has a “signal” value along each IC, we can use the relationship between all 200k “signal” values of two ICs to quantify how much they overlap. Two indices were used to measure the degree of overlap: Pearson correlation and mutual information. The Pearson correlation between the values of any pair of ICs is zero, consistent with the procedure of ICA. The mutual information between the values of any pair of ICs is less than 0.05 (mean value=0.01). If two ICs were completely dependent on each other (or in other words, overlapping), we would expect both indices to be 1. Therefore, the 25 independent components can be considered separable from each other.

We have now added the following text (page 2):

By comparing the “signal” values of different IC dimensions for the 200k images, we found that these dimensions were largely independent of each other (Pearson correlation=0, mutual information<0.05 for all IC pairs).

2. Connected to the previous question, how do I interpret the “positive” and “negative” images for each IC? Do the two groups represent two extremes on an axis (for instance, hypothetically if an IC axis was color, would an image in the positive group contain color and an image in the negative group contain less color, as is also evident in the representative images shown in Fig. S3B2?). Further, if the representative images were extracted using the smallest and largest angle between their IC coordinates and the IC axis, was it possible that some of the images overlapped with other ICs (for example, in the schematic illustration of the IC axis and angle in Figure 1B, the largest angle to IC1 could be close to the smallest angle of IC3).

Yes, the representative images were selected based on the angle between their coordinates and a particular IC axis, and the “positive” and “negative” groups represent two extremes of the axis, as the reviewer explained in the “color” example. It was possible for some of the images to overlap, but this happened quite rarely. Note that the largest angle of all the images to an IC dimension, such as IC1, is close to 180 degrees. In this case, the angles of an IC1’s negative representative image to other IC axes are likely to be close to 90 degrees (cosine angle close to 0). This is because the squared cosine angles of a given image to the 25 dimensions add up to 1, and the fact that the squared cosine angle of 180 degrees is 1 forces the cosine angles for other ICs to be close to 0. Thus, this image is unlikely to be a representative image for other axes, which

require cosine angles close to 1 or -1. In fact, we found that among the 2500 (=50*2*25) representative images, there was only one image that appeared more than once, which is much lower than the chance level—by randomly selecting 25 groups of 100 images, there will be on average 14.9 images that appear more than once (10000 rounds of random sampling).

We have now added (page 3):

Fifty images were selected for each group, resulting in a total of 2500 images (=50 images*2 groups*25 ICs). Among all these images, there was only one image that appeared more than once, which is much lower than the chance level (=14.9 images, estimated by 10000 rounds of randomly selecting 25 groups of 100 images from the 200k images).

3. It says in the methods that 50 representative images were selected for each positive and negative group of each IC, and each image was shown for 500ms, but each block was only 24s long? Does this mean only 48 images were shown? Also, did the positive and negative blocks in each scan contain the exact same images which were repeated? That is, the same block of positive images was shown 8 times and the same block of negative images was shown 6 times in one scan (in the example stimulus sequence in Fig. 1E). If so, was the order of the images within a block randomized each time?

We thank the reviewer for pointing this out. Yes, 48 images were presented in each block, which were randomly selected from the 50 representative images for each block. Therefore, the 48 images were not always the same, but two different blocks of the same group (negative or positive group of the same IC) shared at least 46 images. The difference in the 25D feature between different blocks of the same group is also very small. We averaged the 25D coordinates of the 48 images in each block and compared it with other blocks within the same group, and we found that the cosine angle between blocks is 0.9993 on average. As shown in Figure 1E, the positive image block was shown 8 times and the negative image block was shown 6 times in one scan. We did use the same number of positive and negative blocks for the analysis, i.e., 6 repetitions for both, as indicated by the dashed lines in the current Figure 1E—the first block of the whole scan and the first block after the control and gray screen blocks were not included. The order of the images within a block was randomized each time.

We have now clarified this in the methods (page 24):

In each block, 48 images were randomly selected from the 50 representative images and presented in random order. Each scan consisted of 8 blocks of positive images and 6 blocks of negative images. The same number of positive and negative blocks were used in the analysis (=6 blocks), as indicated by the dashed lines in the current Figure 1E.

We have also indicated this in the figure legends (page 5):

In each block, 48 images were randomly selected from the 50 representative images and presented in random order. The dashed lines indicate the blocks used in the following analyses.

4. It could be useful if the authors indicated exactly where on the surface (Fig. 1F1) corresponds to the flat map (Fig. 1F2).

We thank the reviewer for this suggestion. The region of the inflated surface corresponding to the flat map in Fig. 1F₂ is now indicated by the purple patches in Fig. 1F₁. Note that to better

visualize the full extent of this region, a different perspective of the surface (bottom below) is also shown.

5. In Fig. 2, what do the colors in the IC maps shown in A correspond to? Does close to 1 indicate the response to positive images and close to -1 indicate the response to negative images? This should be made clearer in the legend, as well as the text.

Yes, the reviewer is correct: values close to 1 indicate stronger responses to positive images than to negative images, and values close to -1 indicate the opposite.

We have now clarified this in the legend and in the text (pages 5 and 6):

In these maps, values close to 1 indicate stronger responses to positive images than to negative images, and values close to -1 indicate the opposite.

6. I had some difficulty understanding what exactly “visual features” refer to, and how they were compared with “neural features” in the section “explaining the feature preference map using interpretable visual features”. Do the “visual features” simply refer to the activations of the DNN units, while the “neural features” refer to the neural responses obtained from the fMRI experiments? If not, how exactly were the “visual features” measured/extracted?

Yes, the “neural features” refer to the 25D vectors obtained from the neural responses in the fMRI experiments. The term “visual feature” is used to refer to something a bit more general than the activations of DNN units: a feature is simply a label or an index of an image that reflects some property of the image. For a particular visual feature, such as color, each image used in the experiment was assigned a value related to the feature, depending on a particular property of the image, such as how colorful the image is. To compare these “visual features” with the “neural features”, we “projected” them into the 25D IC space. Typically, this was done as follows: the corresponding feature value was first extracted from each image in the 25-IC stimulus set, and was then averaged across images within “positive” and “negative” blocks of different ICs. The difference between the average feature values of two blocks was computed for each IC, and the resulting 25D vector was normalized to unit length, similar to what we did when converting the fMRI signals of each brain location into a 25D “neural feature”. All details regarding the projection of visual features into the 25D space can be found in the Methods section (*The construction of interpretable features and comparison with neural data*). It may be a bit confusing to refer to the blue feature in the IC space as the “visual feature” in Figure 3B, since it’s actually the projection of the “visual feature” in that space. We have now referred to it as the “projected visual feature”. We have also added the following paragraph to the main text to clarify the meaning of the visual features (page 7):

The term “visual feature” is used to refer to a label or an index of an image that reflects some property of that image. In total, 21 features were examined, including: low-level features, such as energy at specific spatial frequencies and color; mid-level shape features, such as curvature; high-level semantic features, such as animacy. To directly compare these “visual features” with the “neural features” (the 25D preferred feature measured experimentally), we projected them into the 25D IC space (Figure 3B). Typically, this projection was done by first averaging the

corresponding values of the visual feature across images within “positive” and “negative” blocks of different ICs, and then computing the difference between the average feature values of two blocks for each IC and normalizing the 25D vector to unit length (for details, see Methods: *The construction of interpretable features and comparison with neural data*).

7. If this is possible, it could be useful to project the feature map shown in Fig. 4A or Fig. 5A1 on the flat map of the brain with the sulci visible underneath the feature map – this could be helpful in anatomically orienting the reader and understand the location of the feature maps in relation to the different regions in the temporal lobe.

We thank the reviewer for this suggestion. The feature map in Figure 5A₁ has now been overlaid on the flat map with the sulci visible. To avoid overcrowding, the overlays are shown as separate plots in the insets. The boundaries of the sulci have been highlighted for better visibility.

8. In the section “testing new predictions based on the 25D preference maps”, the authors say that positive and negative representative images were selected for the 1st residual feature (or feature 8), which were then used in an fMRI experiment. I assume that this means that the representative images of the 1st residual feature used in this fMRI experiment were selected from the 200k natural images from ImageNet. Were the synthesized images (shown in Fig. S3D3) not used in any experiments?

Yes, the representative images in Figure 5A₂ were selected from the 200k natural images in ImageNet. The synthesized images in the old Figure S3D₃ (Figure S4D₃ in the current version of the manuscript) were not used in the experiments in the last version of the manuscript, because we figured out how to use the generative model only after we had completed most of the recordings. Now we have recorded more neurons in the newly identified region while presenting the synthesized images to the animal. We found that the neurons in this new region (region P in Figure 5C) responded more strongly to the synthesized images of the positive direction of this feature than to those of the negative direction (Figures R4A and B). Interestingly, the positive synthesized images elicited even stronger responses than the positive representative images selected from ImageNet (Figure R4C). We think this is because the synthesized images were optimized to align with the target feature, resulting in a stronger similarity to the target feature than the representative natural images (cosine angle between the 25D coordinates of the images and the target feature: mean±SD=0.87±0.03 for synthesized images, and 0.63±0.03 for natural images).

We have now added the new result as Figure 5G (note that Figures 5F and 5E are also updated since more neurons have been recorded) and the following text in the manuscript (pages 12-13): Finally, in addition to the representative images, we also presented the synthesized images for the same feature to the monkey (Figure S4D₃), and found that neurons in the new region (Figure 5C) responded more strongly to the synthesized images of the positive direction of that feature than to those of the negative direction (Figure 5G₁ and the brown line in 5G₂). Interestingly, the positive synthesized images elicited even stronger responses than the positive representative images selected from ImageNet (black line in Figure 5G₂), likely because the synthesized images are better aligned with the target feature.

We have also clarified in the figure legends for the representative natural images (Figure 5A₂) on page 14: representative images were selected from the 200k natural images in ImageNet.

Figure R4. Response of neurons in the new region to synthesized images.

A) Peristimulus time histogram of an example neuron's responses to the synthesized images of the positive and negative directions of the same feature in Fig. 5A₂ (see Fig. S4D₃), smoothed using a 25-ms sliding window. The neuron was recorded in region P (Figure 5C₁). Shadings represent SEM.

B) Average time courses of t-contrasts between positive and negative conditions for synthesized images (brown; see Fig. S4D₃) and natural images (purple; see Fig. 5A₂), respectively. Shadings represent SEM. Lines on the top indicate the time windows with significant differences from 0 ($p < 0.01$, Student's t-test) for the two stimulus sets. All neurons here were recorded in region P.

C) Same as (B), but for t-contrasts between the positive condition of synthesized images and that of natural images.

References:

- Cadiou, C. F. *et al.* Deep neural networks rival the representation of primate IT cortex for core visual object recognition. *PLoS Comput Biol* **10**, e1003963, doi:10.1371/journal.pcbi.1003963 (2014).
- Schrimpf, M. *et al.* Brain-Score: which artificial neural network for object recognition is most brain-like? *bioRxiv* <https://doi.org/10.1101/407007> (2018).
- Jozwik, K. M., Schrimpf, M., Kanwisher, N. & DiCarlo, J. J. To find better neural network models of human vision, find better neural network models of primate vision. *bioRxiv* <https://doi.org/10.1101/688390> (2019).
- Bao, P., She, L., McGill, M. & Tsao, D. Y. A map of object space in primate inferotemporal cortex. *Nature* **583**, 103-108, doi:10.1038/s41586-020-2350-5 (2020).

Reviewer #3:

The authors collected monkey and human fMRI data in response to images based on the independent components of the representational spaces of late stage Alexnet Model. They conduct numerous complex analyses, and argue their approach “fills-in gaps” of high-dimensional feature maps of primate temporal lobe, and “pinpoints critical differences” in the topographic maps of monkeys, humans, and “AI”.

I found the approach to measure responses in both monkey and humans and explore convergences in their topographies to be a novel contribution. However, I found the manuscript to be not well-situated in the current literature, with overly broad scope, overly complex analyses, and unclear contributions to current understanding of the ventral visual stream topography in primates. I detail these below.

We thank the reviewer for the critical comments, which help us to rethink our results, to connect them with past literature, and to distill what is really novel. We have now substantially revised the manuscript to clarify these points.

1. The work is not situated in the literature well. The work makes claims that are already known, and sets up the problem not recognizing both earlier, and more recent progress.

For example, the motivating set up is: “a panoramic view of topographic organization is still lacking”. This is false. Consider work by Haxby et al., 2011; Huth & Gallant; Konkle & Caramazza 2017; Conway’s 2018 Ann Review; not to mention Bao et al., 2020! There is extensive work charting the large-scale organization of object information and tuning, both empirically and theoretically. As a specific example, consider the title: “high dimensional organization of visual features in primate temporal lobe”. Note Haxby et al., title from 2011 (a decade ago!): “A Common, High-Dimensional Model of the Representational Space in Human Ventral Temporal Cortex.” And Bao et al 2020: “A map of object space in primate inferotemporal cortex”. There are also proposals for a panoramic view of the topographic organization: Konkle & Carmazza, 2013: “Tripartite Organization of the Ventral Stream by Animacy and Object Size.” Even earlier work by Levy, Malack, and Hasson (e.g. Hasson et al., 2004) charted a large-scale map of object cortex (with no gaps!) and presented a theory for its large-scale formation.

Additionally, there are now a number of models that take deep net spaces and map them topographically in different ways (e.g. Blauch et al., Keller & Welling, Doshi & Konkle, Zhang et al.,). These ‘near-neighbor’ papers are also relevant to the claims and theoretical positioning of this work.

We thank the reviewer for pointing this out. We agree with the reviewer that some of the important studies in the past were not adequately acknowledged in the previous version of the paper, and some of the claims were too broad and inaccurate without proper background. We have discussed the literature more thoroughly and raised more specific questions in the introduction section of the manuscript (page 2):

A series of studies conducted in primate brains have revealed multiple ITC subregions specialized for specific object categories or features (Kanwisher et al., 1997, Epstein and Kanwisher, 1997, and others...). It has been suggested that these subregions are organized along coarse functional gradients of visual features such as animacy and object size (Konkle and Oliva, 2012; Konkle & Carmazza, 2013; Srihasam et al., 2014; Konkle & Carmazza, 2017; Conway, 2018; Bao et al., 2020). The large-scale organization of the temporal lobe has also been characterized using data-driven approaches by showing natural stimuli, such as movies, to the subjects (Hasson et al., 2004; Haxby et al., 2011; Huth et al., 2012). More recently, theoretical studies have shown that applying simple spatial constraints to the backbones of neural networks can lead to topographic organizations similar to those observed in the experiments (Lee et al., 2020; Zhang et al., 2021; Keller et al., 2021; Doshi and Konkle, 2022; Blauch et al., 2022). Despite all these achievements, some important questions remain unanswered, for example: 1) The brain regions identified with specialized functions cover only about half of the monkey ITC (Bao et al., 2020), and it's unclear what the rest does; 2) Most studies have been conducted in a single species, either human or a non-human primate species, and a detailed comparison of the ITC's topographic organization between human and monkey brains is still needed.

2. The framing is overly ambitious, (as are the scope of analyses included in the paper, see below). This makes the contributions of the paper difficult to assess and feel unsubstantiated (undermining the tremendous effort and impressive computational work that went into this project!). This comment related to that above, and can be further evidenced by claims like that of the final sentence of the abstract (“pinpointing the critical differences in the functional organization of high-level visual areas between monkey, human, and artificial intelligence”).

We have now gone through the paper and toned down our claims where necessary, including the sentence in the abstract:

These maps also enabled quantitative analyses of the topographic organization of the temporal lobe, demonstrating the coexistence of multiple functional gradients that differ in orientation and spatial scale, and revealing significant differences in the functional organization of high-level visual areas between monkey and human brains.

3. The method of collecting and analyzing the data had some assumptions that gave me pause and which I needed more explanation for their implications on subsequent analyses and results.

3.a. Data analysis: one early choice you made was to look at activations along ICs relative to a control block of 50 images. This confused me, as each one of these 50 images has some value on each IC. By chance you may have be partially controlling away ICs based on this “control” block. Clarifying the logic and implications of this choice would be important—do you really need to do this step?

We thank the reviewer for pointing this out. We used this control block for normalization because the entire experiment was performed over multiple sessions, and the signal strength inevitably varied between sessions, or even between scans within the same session. Because we used an identical set of images in the control block, the response to this block could serve as a reference

to control for the differential effects of intersession variation on different ICs. This is true even if the 50 images were biased toward some of the ICs. To consider an extreme case, suppose the responses of a voxel to 25 ICs were a_1, a_2, \dots, a_{25} , and the response to the control block was the same as IC1 ($= a_1$). After normalization, the responses to 25 ICs become $1, \frac{a_2}{a_1}, \dots, \frac{a_{25}}{a_1}$. In the later analyses, the preferred feature is normalized to unit length, so in this case, the resulting vector is the same before and after normalization ($= \frac{(a_1, a_2, \dots, a_{25})}{\sqrt{a_1^2 + a_2^2 + \dots + a_{25}^2}}$). Therefore, we think this normalization provides an unbiased way to correct for inter-session variations. We also computed the preferred features based on the unnormalized responses, and found that although the neural representations are still consistent across hemispheres and individuals (Figure R5A), the consistency between individual subjects is lower than that based on the normalized responses (Figure R5B; mean=0.64 without normalization and 0.68 with normalization). The increase in representational consistency suggests that the normalization procedure removes noise and yields more reliable estimates of the underlying feature maps, which should be quite consistent across individuals, as suggested by previous studies showing the stable tuning of the primate visual system (Kriegeskorte et al., 2008; Tsao et al., 2008; Bao et al., 2020).

We have now added the analysis as Figures S2G-H and the following text to the manuscript (pages 3 and 5):

We performed the normalization using the control block because the entire experiment was conducted over multiple sessions, and the signal strength inevitably varied between sessions. Since we used an identical set of images in the control block, the response to this block could serve as a reference to control for the differential effects of intersession variation on different ICs. Furthermore, we found that without the normalization of the control block, the consistency between subjects became lower (Figures S2G-H), suggesting that this normalization step removed noise from the data.

Figure R5. Representational similarity analysis of feature maps without normalization.

A) Following the procedure in Figure 2, the consistency of the similarity matrices between hemispheres, individuals, and species was computed using feature maps without normalization by the control block. Same convention as in Figure 2E. B) Consistency between subjects computed using feature maps before and after normalization are plotted against each other. The identity line is shown as a reference. Different colors indicate different combinations of species. For example, blue dots represent the intersubject consistency between three monkeys.

3.b. Data analysis: The images were selected based on the 25 ICs to reflect the positive and negative ends, presented in a blocked design. This means that by definition you will be measuring how these pre-defined axes project into the (stable!) tuning of the object cortex. Of course there will be a 'topography' of these ICs, because each voxel/vertex has some tuning function, which bears some relationship to the ICs, and so you can just make a map of ICs. This is assumed by the experimental design and required by the data analysis pipeline and not testing anything per se. The similarity matrix of the ICs should be similar because there's stable tuning of the visual system... the degree to which they are dissimilar really means you have noisy data, I think? And the claim that you have multiple oriented gradients... I think this is required from your analysis rather than a discovery where alternate patterns of results were possible. Is there any way you could have found something else? I'm not sure and it was hard to follow complex analysis procedures used here.

We agree with the reviewer that the consistency in representational similarity in Figure 2 is expected from the stable tuning of the visual system. This is not a very novel finding per se, but suggests that the procedure in our study can provide reliable estimates of the feature maps, which form the basis of the analyses later in the paper. We have now clarified this (page 5):

It should be noted that the consistency between representational similarities is expected from the stable tuning of high-level visual areas as demonstrated in previous studies (Kriegeskorte et al., 2008; Tsao et al., 2008; Bao et al., 2020), and the purpose of this analysis is to illustrate the reliability of our procedure.

Regarding the oriented gradients, we understand the reviewer's concern as follows: since this high-dimensional feature map is embedded in the 2D cortical surface, it seems expected that different features will display different spatial patterns, and one likely solution is to represent different features with different oriented gradients. If so, are there other the alternative solutions? We would like to emphasize that one of the main purposes of the computational models in Figure 7 and Figure S7 is to provide such alternative solutions. These models are able to embed a high-dimensional representation in a 2D surface and the units are organized under spatial constraints that give rise to known topographic structures such as face patches. Note that different types of control models were constructed: besides the topographic DNN in Figure 7, we also built SOMs (self-organizing maps; Figures S7A-B), similar to previous studies (Zhang et al., 2021; Doshi and Konkle, 2022), as well as a baseline model by randomly shuffling the locations of the monkey feature map while satisfying a certain constraint of spatial continuity (Figures S7C-D). After generating the control maps, we think it's important to use a quantitative approach to analyze the organizational pattern so that the model maps and the brain maps can be directly compared. Therefore, spatial autocorrelation was used to extract several indices from each map, including the orientation selectivity and orthogonality indices in Figure 7, which support that the pair of orthogonally oriented gradients is much weaker in the models than in the monkey brains. We have now rephrased the motivation of the computational modelling section to make it clearer (pages 18-19):

Our analyses suggest the monkey temporal lobe has an intricate organization of visual features, where the spatial scales of functional gradients depend on their orientation. However, since this high-dimensional feature map is embedded in the 2D cortical surface, it seems expected that different feature dimensions will display different spatial patterns, and a likely solution is to

represent different features with different oriented gradients. If this is the case, the reader may ask: are there other alternative solutions under the constraint of spatial continuity? In this section, we will quantitatively compare primate feature maps with several alternative models.

3.c. You also talk about the dimensions of ICs, but only sampled positive/negative poles, so it is unclear whether you have any evidence for dimensional gradients.

We think the reviewer’s concern is that if only the positive/negative poles were sampled, it’s hard to claim that there is a continuous gradient for a given dimension because the full tuning along that dimension is not available (Figure R6A). We understand this concern, but perhaps what we mean by “gradient” was not clearly explained in the previous version of the manuscript, so let’s clarify it first: Our study is based on the assumption that IT neurons linearly encode axes of high-dimensional spaces spanned by DNN features (Cadieu et al., 2014; Bao et al., 2020), so that the tuning of each brain location is represented by a 25D axis in our feature map. The relationship between each location’s tuning and a given feature dimension (such as IC1 or the “animacy” feature) is quantified by the cosine angle between the 25D preferred axis and that dimension: a positive value indicates a positive relationship, a negative value indicates a negative (or opposite) relationship, and a value close to 0 indicates no relationship. In this context, the “gradient” refers to the gradual change in the similarity between the 25D preferred features (the encoded axes) and a given dimension across the cortical surface (Figure R6B). Admittedly, understanding the significance of these gradients requires the assumption of the axis coding scheme, which is a reasonably good approximation of the true tuning of IT neurons (but still an approximation). We agree with the reviewer that in the future we could sample multiple locations along a single axis to reveal the full tuning along that axis, which would better reveal the functional gradients for individual dimensions. We have now added Figure R6B as Figure S6J and the following clarifications (pages 11 and 23):

Here, we used the term “gradient” to indicate the gradual change in the similarity between the 25D preferred features with a given dimension across the cortical surface (Figure S6J).

Our design of sampling the negative and positive ends of each IC dimension was based on the assumption that IT neurons linearly encode axes of high-dimensional spaces spanned by DNN features (Cadieu et al., 2014; Bao et al., 2020). In the future, multiple locations along a single axis could be sampled to reveal the full tuning along that axis, which would better reveal the functional gradients for individual dimensions.

Figure R6. Two ways of defining the functional gradient.

A) Each line indicates the tuning curve of a brain region along a feature dimension. The “gradient” refers to the gradual change in the preferred feature value across the cortical surface. B) Each colored arrow indicates the 25D preferred feature of a brain region. The “gradient” refers to the

gradual change in the angle between the preferred features and a given dimension (black arrow) across the cortical surface. This is the definition we used in the paper.

4. The analysis pipeline was too complex and hard to follow

4.a. You went from 25 ICs (Figure 2), to a map of 21 'intuitive features' (Figure 3c) to a sub-selection of 9 features (Figure 3F), to a set of 3 (Figure 4a). At each step the transformations and assumptions, for me, further obscured insights into the tuning, rather than clarify them. Staying closer to the data, revealing the structure in the maps, and distilling the key insights would clarify the contributions

We thank the reviewer for this comment. We agree that some of the analysis steps could be better explained. The main reason why we chose to compare 25D features with 21 "intuitive features" is that we want to link our feature maps to features that are already known so that novel features represented in the temporal lobe can be identified. Since we could see that some of the 21 features explain the responses much better than other features and some of the weakest features may not contribute significantly to the explaining the neural data (Figure 3C), we think that a feature selection procedure can be performed to identify a set of "visual features" that optimally explain the "neural features". This was done using the half-split approach, resulting in 7 features (Figure 3D). Interestingly, we found that these 7 features explained only ~56% of the neural responses after being normalized by the noise ceiling, suggesting that some unknown features are encoded by the primate temporal lobe. Two additional features were extracted directly from the neural responses (using PCA) after being orthogonalized by the 7 features, resulting in a total of 9 features. We think that these two steps are important to quantitatively link our feature maps to features studied in previous literature and to identify novel features in our maps, but the motivations can be better explained. We agree with the reviewer that the bar plots of three features in Figure 4A are confusing and we have now removed them.

We have now added (pages 7 and 8):

The squared cosine angles (SCA) between neural and visual features were used to quantify how well each interpretable visual feature explained the neural data. This allowed us to quantitatively link our feature maps to already known features and potentially identify novel features represented in the temporal lobe.

Examining the squared cosine angles, we can see that some of the 21 features explain the responses much better than other features. To identify the set of most explanatory features, we designed a feature selection procedure...

Interestingly, we found that these seven features explained only ~56% of the neural responses after being normalized by the noise ceiling, suggesting that some unknown features are encoded by the primate temporal lobe. Next, we extracted additional features from the neural data, by first orthogonalizing the neural features with respect to the top seven features...

4.b. Relatedly, the use of low-, mid-, and high-level features and the logic around how you ruled out mid-level features was faulty. This was not an experimental study designed to tease these levels of representation apart, and I think they confuse the issue. Your study was aimed to see if the ICs of the late stage of an Alexnet give insight into the large-scale organization of monkey and

human object cortex. Focus on that. Delineating what ‘level’ these features are at requires different data (e.g. see Long et al., 2018, Jagadeesh & Gardner, 2022), and is itself a rather thorny issue that I’d recommend not focusing on here.

We agree with the reviewer that pinpointing the level of features encoded by different brain regions requires different data sets. We have basically removed the part regarding the level of representation.

We have removed the claims about the weakness of mid-level features and the section on analyzing the depth of representation using different layers of networks. The terms such as low-level features are retained when we introduce the 21 features, but we have made it clear that this is more a way of intuitively understanding what the features are, rather than resolving the issue of the level of representation in the temporal lobe (page 7):

We use the terms “low-”, “mid-” and “high-level” only to help the reader intuitively understand what the features are—thoroughly addressing the level of neural representation requires different stimulus sets and experimental designs (Long et al., 2018, Jagadeesh & Gardner, 2022).

5. I found the scope of included results was too extensive to vet everything. For example, by the end of the results section, you implemented your own TDANNs to see if “orientation-dependent organization ... naturally emerged in networks trained with orientation-independent spatial loss”-- this felt like an unnecessarily large addition, as there are again many many choices here, and it is very hard follow the logic of which choices you made, and how you assessed it, and how your supported your claims. This could be a whole additional paper! Further, this is not the only topographic deep net model and so claims that these deep nets don’t replicate broader orthogonal gradients are, e.g., not in line with approaches like Zhang et al, and Doshi & Konkle, 2022

We agree with the reviewer that there are many possible choices of topographic network models, and it’s hard to exclude all possible models. On the other hand, we still want to keep this part, but for a different purpose, i.e., to provide alternative models of topographic organization for comparison with the primate feature maps, as we explained in the point 3.b. Note that we have also implemented the self-organizing maps (Figures S7A-B), similar to previous studies (Zhang et al., 2021; Doshi and Konkle, 2022). By quantitatively analyzing the organizational pattern of the model maps and the brain maps, we show that the pair of orthogonally oriented gradients does not appear in the alternative models, or at least not as strongly as in the monkey feature map. We agree that it is impossible to test all possible models, but at least we can say that this pattern of orthogonal gradients is not an inevitable result of embedding a high-dimensional map into the 2D cortical surface. Overall, we think that with some revisions these models can still be an integral part of the paper, but if the reviewer thinks that not all of the models are necessary, we could simplify this section so that fewer models are presented. We have now removed the claims that TDANNs with orientation-independent spatial loss cannot reproduce our pattern, and emphasized that the models we have provided are alternative models that embed a high-dimensional map into the 2D cortical surface under the constraint of spatial continuity. For example, we have added the following to the text (page 20):

Our findings suggest that the intricate spatial organization observed in the monkey brain is not an inevitable result of embedding a high-dimensional map into the 2D cortical surface under the constraint of spatial continuity. Note that since we couldn't test all possible models, it is likely that the pattern in the monkey data could emerge under some other constraints, but we think identifying such constraints is beyond the scope of the current paper.

Since the purpose of Figure 7 is no longer to exclude models that cannot explain the biological feature map and to identify those that can, the models with anisotropic spatial loss have been removed.

6. Take home results unclear. By the end of the results section, I was not left with a clear sense of the key take home points of how my understanding of the object-cortical organization should be updated.

We have now discussed what we think is novel about our study in the first paragraph of the discussion section:

In this paper, we constructed a high-dimensional space of visual features using a deep network previously shown to be a good model of high-level visual neurons, and systematically characterized the functional feature map in both monkey and human brains. We made the following major discoveries: 1) a new functional subdomain encoding a novel feature was identified in the monkey temporal lobe (Figure 3 and Figures 5A-G); 2) the monkey feature map consisted of a pair of orthogonal gradients with different spatial scales, which were significantly less salient in the human brain (Figures 6 and 7; Table 1); 3) the human brain showed a stronger preference for inanimate objects than the monkey brain (Figures 5I-J; Table 1).

In addition, a new table summarizing the similarities (blue) and differences (red) between the two species, the same one as Table R1, has been added to the discussion section (page 22).

Species Property	Monkey	Human
Feature selectivity	Animate region: Strong preference for animals Inanimate region: Weak preference for inanimate objects; strong preference for energy at high spatial frequency	Animate region: Strong preference for animals Inanimate region: Strong preference for inanimate objects; weak preference for energy at high spatial frequency
Topographic organization	High-dimensional feature map; alternating “animacy” and “residual” feature preferred regions are neighbored by “high sf” preferred region A pair of orthogonal gradients with different spatial scales is clearly present	High-dimensional feature map; alternating “animacy” and “residual” feature preferred regions are neighbored by “high sf” preferred region No clear evidence for orthogonal gradients

Table R1. Cross-species comparison

Blue: similarities between two species; Red: differences between two species

Discussion did not appropriately acknowledge prior work. As noted in above, claiming there is ‘no general principle underlying the topographical organization of primate temporal lobe’ fails to acknowledge prior work (c.f. Bao, Konkle, Hasson, Conway, Haxby, Huth). I found the discussion around the level of representation of tuning to be logically unsound.

We have now removed similar claims and rewritten the discussion to appropriately acknowledge prior work. For example, we have added the following to the discussion section (page 22):

Over the past decades, the functional organization of high-level visual areas in the primate brain has been intensively investigated. Past studies have identified several networks specialized for processing a particular type of object or feature (Kanwisher et al., 1997, Beauchamp et al., 1997, and others...), revealed the large-scale organization of high-level visual areas (Hasson et al., 2004; Haxby et al., 2011; Huth et al., 2012), and proposed general principles underlying such organization (Konkle and Oliva, 2012; Konkle & Carmazza, 2013; Srihasam et al., 2014; Konkle & Carmazza, 2017; Conway, 2018; Bao et al., 2020).

The section related to the level of representation has been removed.

Despite these critics, I feel there are real areas of novelty that could be distilled and treated with a more careful curated scope and lead to an important, high-impact contribution. For example, this is, to my knowledge, once of the very few condition-rich datasets collected in both humans and monkeys, and I don't know of any careful topographic work comparing their spatial organizations. Further, you highlight that your data reveal fine-grained differences in how humans and monkeys represent in animate objects, and in their topography. Clarifying these empirical results, simply and clearly, would be an important and--to my knowledge--novel contribution.

We thank the reviewer for recognizing the novelty of the paper despite the critics. As mentioned above, we have now rewritten the paper to remove unnecessary sections and emphasize the novelty of our findings, including the addition of a new table that summarizes the comparison of the two species.

Further, clarifying that you're using the ICs of an Alexnet as a lens into the tuning is going to be important in how you scope your claims. For example, it is important to acknowledge and discuss the degree to which your claims/results depend on their being a good alignment between these features spaces, and what happens if these ICs are not in good alignment with the true underlying tuning functions of the ventral visual stream voxels/vertices.

Yes, we agree. We have now added the following to clarify this point (pages 22-23):

By analyzing the neural tuning using features derived from AlexNet, what we have done is like projecting the true neural tuning into a subspace spanned by AlexNet features (Figure S1E). Although AlexNet has been shown to be an excellent model of high-level visual areas (Cadieu et al., 2014; Bao et al., 2020; Chang et al., 2021), there is still likely to be a significant fraction of neural responses that cannot be explained by these features. Future work is needed to explore

additional features to approach the true tuning functions.

Overall, it is clear that the authors have carried out an impressive amount of work, both in the dataset collection and in the analytical approaches. There is likely more than one paper here. However, as currently written, scoped, and situated, the novel contributions of the work are obscured and unsubstantiated.

Once again, we thank the reviewer for the careful reading and critical comments. We believe that the paper has been significantly improved by incorporating the revisions mentioned above.

References:

- Kanwisher, N., McDermott, J. & Chun, M. M. The fusiform face area: a module in human extrastriate cortex specialized for face perception. *J Neurosci* **17**, 4302-4311 (1997).
- Epstein, R. & Kanwisher, N. A cortical representation of the local visual environment. *Nature* **392**, 598-601, doi:10.1038/33402 (1998).
- Konkle, T. & Oliva, A. A real-world size organization of object responses in occipitotemporal cortex. *Neuron* **74**, 1114-1124, doi:10.1016/j.neuron.2012.04.036 (2012).
- Konkle, T. & Caramazza, A. Tripartite organization of the ventral stream by animacy and object size. *J Neurosci* **33**, 10235-10242, doi:10.1523/JNEUROSCI.0983-13.2013 (2013).
- Srihasam, K., Vincent, J. L. & Livingstone, M. S. Novel domain formation reveals proto-architecture in inferotemporal cortex. *Nat Neurosci* **17**, 1776-1783, doi:10.1038/nn.3855 (2014).
- Konkle, T. & Caramazza, A. The Large-Scale Organization of Object-Responsive Cortex Is Reflected in Resting-State Network Architecture. *Cereb Cortex* **27**, 4933-4945, doi:10.1093/cercor/bhw287 (2017).
- Conway, B. R. The Organization and Operation of Inferior Temporal Cortex. *Annu Rev Vis Sci* **4**, 381-402, doi:10.1146/annurev-vision-091517-034202 (2018).
- Bao, P., She, L., McGill, M. & Tsao, D. Y. A map of object space in primate inferotemporal cortex. *Nature* **583**, 103-108, doi:10.1038/s41586-020-2350-5 (2020).
- Hasson, U., Nir, Y., Levy, I., Fuhrmann, G. & Malach, R. Intersubject synchronization of cortical activity during natural vision. *Science* **303**, 1634-1640, doi:10.1126/science.1089506 (2004).
- Haxby, J. V. *et al.* A common, high-dimensional model of the representational space in human ventral temporal cortex. *Neuron* **72**, 404-416, doi:10.1016/j.neuron.2011.08.026 (2011).
- Huth, A. G., Nishimoto, S., Vu, A. T. & Gallant, J. L. A continuous semantic space describes the representation of thousands of object and action categories across the human brain. *Neuron* **76**, 1210-1224, doi:10.1016/j.neuron.2012.10.014 (2012).
- Lee, H. *et al.* Topographic deep artificial neural networks reproduce the hallmarks of the primate inferior temporal cortex face processing network. *bioRxiv* <https://doi.org/10.1101/2020.07.09.185116> (2020).
- Zhang, Y., Zhou, K., Bao, P. & Liu, J. Principles governing the topological organization of object selectivities in the ventral temporal cortex. *bioRxiv* <https://doi.org/10.1101/2021.09.15.460220> (2021).
- Keller, T. A., Gao, Q. & Welling, M. Modeling category-selective cortical regions with topographic variational autoencoders. *arXiv* <https://arxiv.org/pdf/2110.13911.pdf> (2021).

Doshi, F. R. & Konkle, T. Visual object topographic motifs emerge from self-organization of a unified representational space. *bioRxiv* <https://doi.org/10.1101/2022.09.06.506403> (2022).

Blauch, N. M., Behrmann, M. & Plaut, D. C. A connectivity-constrained computational account of topographic organization in primate high-level visual cortex. *Proc Natl Acad Sci U S A* **119**, doi:10.1073/pnas.2112566119 (2022).

Kriegeskorte, N. *et al.* Matching categorical object representations in inferior temporal cortex of man and monkey. *Neuron* **60**, 1126-1141, doi:10.1016/j.neuron.2008.10.043 (2008).

Tsao, D. Y., Moeller, S. & Freiwald, W. A. Comparing face patch systems in macaques and humans. *Proc Natl Acad Sci U S A* **105**, 19514-19519, doi:10.1073/pnas.0809662105 (2008).

Long, B., Yu, C. P. & Konkle, T. Mid-level visual features underlie the high-level categorical organization of the ventral stream. *Proc Natl Acad Sci U S A* **115**, E9015-E9024, doi:10.1073/pnas.1719616115 (2018).

Jagadeesh, A. V. & Gardner, J. L. Texture-like representation of objects in human visual cortex. *Proc Natl Acad Sci U S A* **119**, e2115302119, doi:10.1073/pnas.2115302119 (2022).

Sedigh-Sarvestani, M. *et al.* A sinusoidal transformation of the visual field is the basis for periodic maps in area V2. *Neuron* **109**, 4068-4079 e4066, doi:10.1016/j.neuron.2021.09.053 (2021).

Cadiou, C. F. *et al.* Deep neural networks rival the representation of primate IT cortex for core visual object recognition. *PLoS Comput Biol* **10**, e1003963, doi:10.1371/journal.pcbi.1003963 (2014).

Chang, L., Egger, B., Vetter, T. & Tsao, D. Y. Explaining face representation in the primate brain using different computational models. *Curr Biol* **31**, 2785-2795 e2784, doi:10.1016/j.cub.2021.04.014 (2021).

Reviewer #1 (Remarks to the Author):

The authors addressed all my concerns. I think this is a great contribution to the field.

Reviewer #2 (Remarks to the Author):

The revised ms addresses all my concerns.

Reviewer #3 (Remarks to the Author):

The authors have done a thorough and systematic job addressing my comments. (I also appreciate the new clarity detailing the differences between monkey and human topographies.)

I have only a few remaining comments/suggestions.

1- One of the main stated findings from the abstract is: "demonstrating the coexistence of multiple functional gradients that differ in orientation and spatial scale." First, I think you mean "feature gradients" or "feature tuning gradients"; I don't know what you mean by "functional" gradients here, and I think that word is not necessarily justified. Second, the claim about the "coexistence of multiple functional gradients" is stated as a demonstration of an underlying truth of the brain's organization. However, as I read through the analytical approach and results, the critical voice in me wondered: are "multiple feature tuning gradients that differ in orientation in scale" basically a necessary consequence of your analysis pipelines choices? Or, are a they discovery of one whether finding possibility from among many competing possible organizations which could have been found? If the latter, spelling out these alternative out that would be valuable—What other topographic organizations are possible? A set of gradients that were all aligned but vary in the spatial periodicity? A set of gradients all at the same spatial scale but different orientations? a collection of localized patches that don't have periodic spatial structure? For example, you do PCA over the data from the whole cortical population, then project those PCs onto the map. If you had done, say, a non-negative sparse embedding model over that same data, would you still find co-existing functional gradients or would you find little separate clustered regions? Or, perhaps different analysis pipelines more or less appropriate to characterize the true underlying form? At stake with this question is an understanding of your claim of "co-existence of functional gradients." If these gradients are more linked to your analytical approach for how you chose to characterize the tuning, then I'd suggest changing the abstract to that effect (e.g. "we characterize the tuning in terms of multiple feature gradients at different orientations and scales"). If not, a discussion paragraph addressing this potential concern would be valuable.

2- Minor, but valuable for people who are really interested in the human-monkey topography comparisons (me!). Figure 1 shows the inflated monkey brain and the sector of ITC. It would be useful to show the same thing for the human brain and the sector of OTC used—particularly to help orient readers interested in the comparison. (e.g. Figure 3 F,G,H,I show the schematized part of the cortex for both species but it is difficult to orient to the human brain results without seeing the inflated cortical map to know where/how you defined this region.) In general, it seems like your figures mostly show monkey cortex (I think?). I found myself wishing I could see the human and monkey brains next to each other in some of these figures (e.g. Figure 4, 6, 7, supplementary figures). These main figures are all pretty dense already so adding to them may be the move here? Maybe you'd consider making a targeted supplementary figure that has a few more side-by-side comparison of monkey and human organization, over a few of the key analysis/results.

3- In truth, I still found the manuscript result sections to be fairly dense (as well as the figures), with very long paragraphs with intermixed results and motivations and claims and more results. Adding paragraph breaks here and there to break out the different ideas more could help make the work more digestible. Also, it is an impressive amount of work, and that really comes through.

We thank all three reviewers for their careful reading of our paper and their excellent suggestions in both rounds of reviews. By addressing the concerns raised and incorporating our responses into the manuscript, we believe the paper has been significantly improved. Below, we will address each of the specific points raised by the reviewer in detail, with the original comments in blue, our responses in black, and the text added to the manuscript underlined.

Reviewer #1 (Remarks to the Author):

The authors addressed all my concerns. I think this is a great contribution to the field.

Reviewer #2 (Remarks to the Author):

The revised ms addresses all my concerns.

Again, we thank both reviewers for appreciating our work and providing excellent suggestions in the last round of review.

Reviewer #3 (Remarks to the Author):

The authors have done a thorough and systematic job addressing my comments. (I also appreciate the new clarity detailing the differences between monkey and human topographies.)

I have only a few remaining comments/suggestions.

We thank the reviewer for the careful reading of our paper and the insightful comments.

1- One of the main stated findings from the abstract is: “demonstrating the coexistence of multiple functional gradients that differ in orientation and spatial scale.” First, I think you mean “feature gradients” or “feature tuning gradients”; I don’t know what you mean by “functional” gradients here, and I think that word is not necessarily justified. Second, the claim about the “coexistence of multiple functional gradients” is stated as a demonstration of an underlying truth of the brain’s organization. However, as I read through the analytical approach and results, the critical voice in me wondered: are “multiple feature tuning gradients that differ in orientation in scale” basically a necessary consequence of your analysis pipelines choices? Or, are a they discovery of one whether finding possibility from among many competing possible organizations which could have been found? If the latter, spelling out these alternative out that would be valuable—What other topographic organizations are possible? A set of gradients that were all aligned but vary in the spatial periodicity? A set of gradients all at the same spatial scale but different orientations? a collection of localized patches that don’t have periodic spatial structure? For example, you do PCA over the data from the whole cortical population, then project those PCs onto the map. If you had done, say, a non-negative sparse embedding model over that same data, would you still find co-existing functional gradients or would you find little separate clustered regions? Or, perhaps different analysis pipelines more or less appropriate to characterize the true underlying form? At stake with this question is an understanding of your

claim of “co-existence of functional gradients.” If these gradients are more linked to your analytical approach for how you chose to characterize the tuning, then I’d suggest changing the abstract to that effect (e.g. “we characterize the tuning in terms of multiple feature gradients at different orientations and scales”). If not, a discussion paragraph addressing this potential concern would be valuable.

We agree with the reviewer that the term “functional” is ambiguous, and have now replaced “functional gradient” with “feature gradient” throughout the paper.

We thank the reviewer for the critical thought regarding the topographic organization. We agree that procedures such as PCA may favour the presence of periodic patterns in individual components (Dordek et al., 2016), but note that we have also performed the analyses on the original feature maps that were measured experimentally without performing PCA. In old Figures S6F-G (current Figures S7g-j), the low-frequency magnitude of each individual feature map (IC map) is positively correlated with its absolute loading for PC1, but not for PC2, while the high-frequency magnitude showed the opposite pattern. In old Figure 7 (current Figure 9), the analyses were first conducted on individual IC maps, and the resulting polar maps were averaged across the 25 dimensions (current Figure 9e). These averaged maps were then compared between the monkey/human data and alternative models. Thus, the difference we found is unlikely to be a byproduct of PCA. How can we intuitively understand this difference? We compared the monkey feature map with network models at several steps of the analyses (Figure R1; the 1st column = monkey data). While the original map of IC1 for the monkey shows a stripe-like structure (i.e., an oriented gradient), the model maps consist of positive and negative patches that are not aligned along the same orientation (1st row of Figure R1). The autocorrelations of the models are grid-like, quite different from the oriented stripes for the monkey (the 2nd row). Polar maps, as shown in new Figures 9d-e, were computed based on the autocorrelation maps (the 3rd row): while the monkey map for IC1 has a single peak at a high frequency, there are usually more than one peaks in the models, suggesting weaker orientation selectivity in the model gradients. We also examined another dimension (IC17) where the monkey map has a single peak at a low frequency (the 4th row). The polar maps of the models for IC17 have peaks at similar frequencies to the IC1 maps of the corresponding models, but with different combinations of orientations (e.g., the gray circles in the 2nd column, 3rd-4th row, match well with the two peaks for IC1, but not with the peaks for IC17, although they are at the same locations). When 25 maps are averaged, the monkey maps show two peaks with orthogonal orientations at different frequencies, but the model maps show multiple peaks at approximately the same frequency, with ring-like structures (the 5th row). Overall, the network models provide reasonable alternatives: each gradient is periodic but not clearly oriented, with grid-like autocorrelations; different gradients occupy roughly the same spatial scale but are not aligned in orientation. The pair of orthogonal gradients at different spatial scales in the monkey map clearly deviates from this scheme (current Figures 9p-q quantifies this difference). Returning to the sentence in the abstract mentioned by the reviewer, we feel that the statement “the coexistence of multiple feature tuning gradients that differ in orientation in scale” is a bit too general, and have made it more specific: “the existence of a pair of orthogonal gradients that differ in spatial scale”.

We have now added Figure R1 as Figure S10 and the following text to the Discussion:

In Figure 8, the autocorrelation analysis was performed on the projection maps derived from PCA. Previous studies have suggested that PCA may favor the presence of periodic patterns in individual components (Dordek et al., 2016), but note that similar results can be obtained by directly analyzing the original feature maps without performing PCA (Figures S7g-j; Figure 9). To intuitively understand the difference between monkey brains and alternative models, we compared their feature maps at several steps of the analyses (Figure S10). We found that for the network models, each gradient is periodic but not clearly oriented, with grid-like autocorrelations; different gradients occupy roughly the same spatial scale but are not aligned in orientation. These observations are inconsistent with the pair of orthogonal gradients at different spatial scales in the monkey map.

Figure R1. Comparison of monkey and model feature maps across multiple analysis steps

Each column indicates either the monkey or one of the network models in Figures 9f-q. Each row indicates an analysis step, from the original feature map to the autocorrelation map to the polar map as in Figure 9d. Maps in the top two rows are normalized so that the maximum absolute value is 1. The polar maps for IC1 and IC17 are shown, together with the average maps of 25 ICs. The gray circles in the 2nd column, 3rd row, outline the two peaks of the polar map. The same circles do not match well with the peaks of the map below in the 4th row. Scale bar: 1 cm.

2- Minor, but valuable for people who are really interested in the human-monkey topography comparisons (me!). Figure 1 shows the inflated monkey brain and the sector of ITC. It would be useful to show the same thing for the human brain and the sector of OTC used—particularly to

help orient readers interested in the comparison. (e.g. Figure 3 F,G,H,I show the schematized part of the cortex for both species but it is difficult to orient to the human brain results without seeing the inflated cortical map to know where/how you defined this region.) In general, it seems like your figures mostly show monkey cortex (I think?). I found myself wishing I could see the human and monkey brains next to each other in some of these figures (e.g. Figure 4, 6, 7, supplementary figures). These main figures are all pretty dense already so adding to them may be the move here? Maybe you'd consider making a targeted supplementary figure that has a few more side-by-side comparison of monkey and human organization, over a few of the key analysis/results.

We thank the reviewer for this suggestion. The sector of OTC used in our study is now shown on the inflated surface for a human subject as Figure S6. In Figure S11, we have compared several key results between human and monkey brains, including: the projection of feature maps onto the “high sf” feature, the “animacy” feature, and the residual feature (Figure R2a); the clustering analysis (Figure R2b); the autocorrelation analysis on individual IC maps (Figure R2c).

Figure R2. Additional comparisons of monkey and human feature maps.

a) The average feature maps of three monkeys (top) and four human subjects (bottom) were projected onto the “high sf” feature, the “animacy” feature, and the residual feature. The presence of negative projections onto the “animacy” feature for parts of the human map, but not the monkey map, is consistent with the stronger preference for inanimate objects in the human brain. b) Left: The clustering result for the average monkey feature map. Bayesian information criterion revealed the optimal number of clusters to be 15. Same convention as in Figure 5. Right: The clustering result for the average human feature map. The same number of clusters as in the monkey map was used for the human map. While blue and green stripes are visible in the monkey map, there is no such clear pattern in the human map. c) The original feature map, the autocorrelation map, and the polar map as in Figure 9d are shown for IC1 and IC17 of the monkey (top) and the human (bottom) feature maps. The average polar maps of all 25 ICs are also shown. Feature maps and autocorrelation maps are normalized so that the maximum absolute value is 1. While the two ICs peak at different spatial frequencies in the monkey polar map, the two human maps peak at similar frequencies. Scale bars: 1 cm.

3- In truth, I still found the manuscript result sections to be fairly dense (as well as the figures), with very long paragraphs with intermixed results and motivations and claims and more results. Adding paragraph breaks here and there to break out the different ideas more could help make the work more digestible. Also, it is an impressive amount of work, and that really comes through.

We thank the reviewer for this suggestion. Multiple paragraph breaks have been introduced in the sections related to old Figures 2, 3, 5, 6, and 7 (current Figures 2-4, 6, 8-9). Along the same lines, two main figures have been divided into four smaller figures, as suggested by the editor.

Reference

Dordek, Y., Soudry, D., Meir, R. & Derdikman, D. Extracting grid cell characteristics from place cell inputs using non-negative principal component analysis. *Elife* **5**, e10094, doi:10.7554/eLife.10094 (2016).